# Fair Graph Machine Learning under Adversarial Missingness Processes

**Debolina Halder Lina**
Department of Computer Science, Rice University, Houston, TX 77005, USA

**Arlei Silva**
Department of Computer Science & Ken Kennedy Institute, Rice University, Houston, TX 77005, USA

## Abstract

Graph Neural Networks (GNNs) have achieved state-of-the-art results in many relevant tasks where decisions might disproportionately impact specific communities. However, existing work on fair GNNs often assumes that either sensitive attributes are fully observed or they are missing completely at random. We show that an adversarial missingness process can inadvertently disguise a fair model through the imputation, leading the model to overestimate the fairness of its predictions. We address this challenge by proposing Better Fair than Sorry (BFtS), a fair missing data imputation model for sensitive attributes. The key principle behind BFtS is that imputations should approximate the worst-case scenario for fairness—i.e., when optimizing fairness is the hardest. We implement this idea using a 3-player adversarial scheme where two adversaries collaborate against a GNN classifier, and the classifier minimizes the maximum bias. Experiments using synthetic and real datasets show that BFtS often achieves a better fairness $\times$ accuracy trade-off than existing alternatives under an adversarial missingness process.

## 1 Introduction

With the increasing popularity of machine learning in high-stakes decision-making, it has become a consensus that these models carry implicit biases that should be addressed to improve the fairness of algorithmic decisions (Ghallab, 2019). The disparate treatment of such models towards African Americans and women has been illustrated in the well-documented COMPAS (Angwin et al., 2022) and Apple credit card (Vigdor, 2019) cases, respectively. While there has been extensive research on fair ML, the proposed solutions have mostly disregarded important challenges that arise in real-world settings. For instance, in many applications, data can be naturally modeled as graphs (or networks) (Dong et al., 2023), representing different objects, their relationships, and attributes, instead of as sequences or images. Moreover, fair ML is also prone to missing data (Little & Rubin, 2019). This is particularly critical because fair algorithms often require knowledge of sensitive attributes that are more likely to be missing due to biases in the collection process or privacy concerns. For instance, census and health surveys exhibit missingness correlated with gender, age, and race (O'Hare & O'Hare, 2019; Weber et al., 2021). Networked data, such as disease transmission studies, face similar issues (Ghani et al., 1998). As a consequence, fair ML methods often rely on missing data imputation, which can introduce errors that compromise fairness (Mansoor et al., 2022; Jeong et al., 2022).

This work investigates the impact of adversarial sensitive value missingness processes on fairness, where the pattern of missing sensitive data is structured to obscure true disparities. If the predictions from the imputation method fail to capture the true distribution of a sensitive attribute, any fairness-aware model trained on the imputed data is prone to inheriting the underlying true bias. More specifically, a key challenge arises when an adversarial missingness process makes the imputed dataset appear to be fair. In graphs, an adversary can exploit the graph structure to manipulate the imputed values and, as a consequence, the fairness-aware model. Prior work on the fairness of graph machine learning assumes sensitive values are Missing Completely At Random (MCAR). However, this assumption rarely holds in practice (O'Hare & O'Hare, 2019; Weber et al., 2021; Ghani et al., 1998; Jeong et al., 2022). Luh (2022) and Fukuchi et al. (2020) provide motivational examples of how

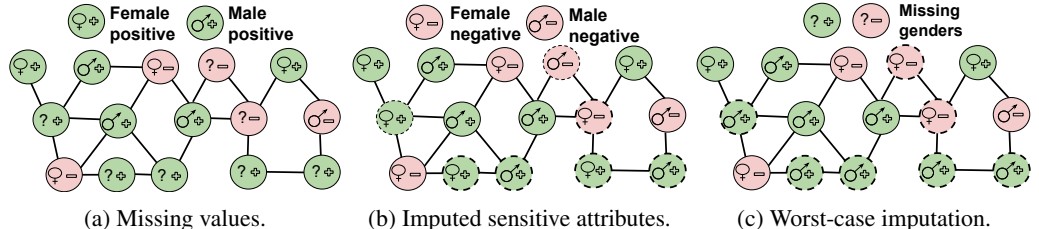

(a) Missing values.     (b) Imputed sensitive attributes.     (c) Worst-case imputation.

Figure 1: In Fig. (a), a graph machine learning algorithm is applied to decide who receives credit (positive or negative) based on a possibly missing sensitive attribute, gender (and binary only for illustrative purposes). As shown in Fig. (b), traditional missing data imputation does not account for outcomes (positive/negative), and thus, their imputed values can under-represent the bias of the complete dataset—demographic parity (DP) is $0.09$ in this example (DP and bias are inversely related). This paper proposes BFtS, an imputation method for graph data that optimizes fairness in the worst-case imputation scenario using adversarial learning, as shown in Fig. (c), where DP is $0.47$.

fairness can be manipulated by adversarial data collection. Methods that overlook the adversarial missingness process risk producing misleading fairness guarantees w.r.t. the complete data.

Figure 1 motivates our problem setting using a toy example. A machine learning algorithm is applied to decide whether individuals should or should not receive credit. Figure 1a shows both the gender and outcome for each individual. The genders of some samples are unknown in Figure 1a, and the demographic parity ($\Delta DP$) considering only the observed values is $0.25$ ($\Delta DP$ is a bias metric described in Section 5). We illustrate how different missing data imputations affect the fairness of credit decisions in Figures 1b and 1c. A straightforward imputation is demonstrated in Figure 1b, where the gender of the majority of neighbors is assigned to the missing attribute, resulting in a best-case scenario in terms of fairness with a $\Delta DP$ of $0.09$. However, in the worst-case scenario, shown in Figure 1c, $\Delta DP$ is $0.47$. If an adversarial missingness process can induce the imputation model to generate the imputation from Figure 1b but the complete data is Figure 1c, a fair model trained on the imputed data will still be biased w.r.t. the complete data.

To counter an adversarial missingness process for sensitive values on graphs, we propose Better Fair than Sorry (BFtS), a 3-player adversarial learning framework for missing sensitive value imputation based on Graph Neural Networks (GNNs). Our experiments show that BFtS achieves a better accuracy $\times$ fairness trade-off than existing approaches, especially under adversarial missing sensitive values.

We summarize our contributions as follows: (1) We investigate theoretically and empirically the potential of an adversarial missingness process to bias a fair GNN; (2) We propose Better Fair than Sorry (BFtS), a novel 3-player adversarial learning framework for the imputation of missing sensitive data that produces worst-case imputed values for fair GNNs and is effective under adversarial missingness processes and even when sensitive attribute information is completely unavailable; and (3) We show empirically that BFtS achieves a better fairness $\times$ accuracy trade-off than the baselines.

## 1.1 RELATED WORK

**Fairness in graph machine learning.** Our work is focused on group fairness Pessach & Shmueli (2022); Dong et al. (2023). Existing work can be grouped into pre-processing, extended objectives, and adversarial learning. Pre-processing methods such as FairOT, FairDrop, FairSIN, EDITS, and Graphair remove bias from the graph before training (Laclau et al., 2021; Spinelli et al., 2021; Yang et al., 2024; Dong et al., 2022; Ling et al., 2023). Objective-based methods—including Fairwalk, Crosswalk, Debayes, MONET, NIFTY, FairVGNN, PFR-AX, FairSAD, and FairGAE—modify GNN losses to learn fair representations (Rahman et al., 2019; Khajehnejad et al., 2022; Buyl & De Bie, 2020; Palowitch & Perozzi, 2020; Agarwal et al., 2021; Wang et al., 2022; Merchant & Castillo, 2023; Zhu et al., 2024; Fan et al., 2021). Adversarial methods such as CFC, FLIP, DKGE, and Debias jointly train GNNs with adversaries for fair prediction (Bose & Hamilton, 2019; Masrour et al., 2020; Arduini et al., 2020; Zhang et al., 2018). All these require fully observed sensitive attributes.

**Missing Data Imputation and fairness.** Missing data can be imputed using unconditional mean, reconstruction, preferential attachment, autoencoders, etc. (Donders et al., 2006; Pereira et al., 2020;

Huisman, 2009). The traditional procedure for handling missing data is independent imputation—i.e., to first impute the missing values and then solve the task (Rossi et al., 2022; Buck, 1960). SAT is a matching-based GNN for graphs with missing attributes (Chen et al., 2020), but it does not account for fairness. Training a classifier from imputed data can amplify the bias of a machine learning model, as discussed in (Subramonian et al., 2022; Zhang & Long, 2021; Feng et al., 2023; Guha et al., 2024; Martínez-Plumed et al., 2019; Fricke, 2020; Mansoor et al., 2022). Some studies try to generate fair feature imputations (Subramonian et al., 2022; Feng et al., 2023; Jeong et al., 2022; Zhang & Long, 2022). However, as the approaches discussed earlier, these methods do not consider missing sensitive information and require full knowledge of the sensitive attributes for fairness intervention.

**Fairness with missing sensitive attributes.** One extreme assumption is the complete unavailability of sensitive information. For example, Hashimoto et al. (2018) considers the worst-case distribution over group sizes, Lahoti et al. (2020) reweighs training samples adversarially, Chai & Wang (2022) minimizes a top-$k$ average loss, and Zhao et al. (2022) reduces correlation between predictions and features associated with sensitive attributes. Other approaches include class re-balancing (Yan et al., 2020) and learning soft labels from an overfitted teacher (Chai et al., 2022). In practice, partial sensitive information is often available and improves fairness (Chai & Wang, 2022), but these models fail to leverage it. FairGNN (Dai & Wang, 2021) assumes limited sensitive data and imputes missing values independently, FairAC (Guo et al., 2023) uses only observed attributes without imputation, and RNF (Du et al., 2021) generates proxy annotations using generalized cross-entropy (Zhang & Sabuncu, 2018). Similar to RNF, our approach handles completely or partially missing sensitive information. BFtS applies a 3-player scheme to minimize the maximum possible bias and outperforms FairGNN and RNF in terms of fairness × accuracy. Minimizing the maximum instead of the average risk, similar to our method, has been shown to achieve better guarantees (Shalev-Shwartz & Wexler, 2016). A similar minmax approach has been applied to maximize the robustness and accuracy of uncertainty models (Löfberg, 2003; Lanckriet et al., 2002; Chen et al., 2017; Fauß et al., 2021). In (Nguyen et al., 2017; Vandenhende et al., 2019), a 3-player adversarial network is proposed to improve the classification and stability of adversarial learning.

**Adversarial attacks on fairness.** Recent work has investigated data poisoning as a means to adversarially degrade model fairness (Mehrabi et al., 2021; Solans et al., 2020). UnfairTrojan and TrojFair introduce backdoor attacks specifically aimed at reducing model fairness (Furth et al., 2024; Xue et al., 2023). In the context of graph neural networks (GNNs), most adversarial fairness attacks modify graph topology: Hussain et al. (2022); Zhang et al. (2024) perturb edges, NIFA injects nodes via uncertainty maximization and homophily enhancement (Luo et al., 2024), and FATE applies bilevel meta-learning for poisoning (Kang et al., 2024). Thus, prior work has primarily focused on poisoning, node injection, or structural perturbations, while attacks that compromise fairness solely by manipulating sensitive-attribute missingness remain unexplored. In this study, we focus on adversarial missingness rather than data poisoning, as it represents a more practical and plausible threat model in many real-world systems. Prior work on adversarial missingness in causal structure learning (Koyuncu et al., 2023; 2024) highlights that when data authenticity is enforced (e.g., via cryptographically signed sensor records), an adversary cannot modify values or inject fabricated samples without violating digital-signature constraints. Such tampering would introduce inconsistencies across logs, backups, and signatures, making it infeasible under standard integrity guarantees. In contrast, selectively withholding, dropping, or failing to log certain fields does not break these integrity checks and is indistinguishable from common pipeline failures. Under this threat model, the adversary is therefore limited to partially concealing existing data and can strategically choose which sensitive attributes to withhold. This makes adversarial missingness a plausible and practically relevant mechanism for undermining fairness.

## 2 PRELIMINARIES

Let $\mathcal{G} = (\mathcal{V}, \mathcal{E}, \mathcal{X}, \mathcal{S})$ be an undirected graph where $\mathcal{V}$ is the set of nodes, $\mathcal{E} \subseteq \mathcal{V} \times \mathcal{V}$ is the set of edges, $\mathcal{X}$ are node attributes, and $\mathcal{S}$ is the set of sensitive attributes. The matrix $A \in \mathbb{R}^{N \times N}$ is the adjacency matrix of $\mathcal{G}$ where $A_{uv} = 1$ if there is an edge between $u$ and $v$ and $A_{uv} = 0$, otherwise. We focus on the setting where sensitive attributes might be missing, and only $\mathcal{V}_S \subseteq \mathcal{V}$ nodes include the information of the sensitive attribute $s_v$ for a node $v$. The sensitive attribute forms two groups, which are often called sensitive ($s_v = 1$) and non-sensitive ($s_v = 0$).

While our work can be generalized to other fairness-aware tasks, we will focus on binary fair node classification, where the goal is to learn a classifier $f_C$ to predict node labels $y_v \in \{0, 1\}$ based on a training set $\mathcal{V}_L \subseteq \mathcal{V}$. Without loss of generality, we assume that the class $y = 1$ is the desired one (e.g., receive credit or bail). Given a classification loss $\mathcal{L}_{class}$ and a group fairness loss $\mathcal{L}_{bias}$, the goal is to learn the parameters $\theta_{class}$ of $f_{class}$ by minimizing their combination:

$$\theta^*_{class} = \arg\min_{\theta_{class}} \mathcal{L}_{class} + \alpha\mathcal{L}_{bias}$$

where $\mathcal{L}_{bias}$ measures the impact of the sensitive attribute $s_v$ over the predictions from $f_{class}$ and $\alpha$ is a hyperparameter. The main challenge addressed in this paper is how to handle missing sensitive attributes. In this scenario, one can apply an independent imputation model $s_v \approx f_{imp}(v)$ before training the fair classifier. However, the fairness of the resulting classifier will be highly dependent on the accuracy of $f_{imp}$, as will be discussed in more detail in the next section.

## 3 INTRODUCING BIAS VIA AN ADVERSARIAL MISSINGNESS PROCESS

We investigate the impact of missing sensitive values on fairness in graph machine learning. Datasets with substantial missingness typically require imputation. In this section, we focus on how the missingness process (i.e., the process that generates missing values) can lead to (intended or unintended) biases in a fair model trained using the imputed data, leading fair models to underestimate bias and remain unfair relative to the complete data.

We can describe the missingness process using a threat model. The asset is the graph $\mathcal{G}(\mathcal{V}, \mathcal{E}, \mathcal{X}, \mathcal{S})$, and the threat is increasing the bias of a node classification model $f_{class}$ applied to $\mathcal{G}$. The vulnerability arises because the adversary can select a subset of sensitive attributes $\mathcal{V} \setminus \mathcal{V}_S$ to be missing, thereby inducing a strategically biased missingness pattern. Our mitigation combines missing-data imputation and fair node classification to counter such adversarial missingness. The attacker's capabilities and constraints are described next in the adversarial missingness formulations introduced in this section.

The simplest missingness process for sensitive values is *missing completely at random*—i.e., the probability of a value being missing is independent of the data. However, in practical scenarios, missing values are rarely random (O'Hare & O'Hare, 2019; Weber et al., 2021; Ghani et al., 1998). We focus on the particular case where the missingness process is adversarial. We prove that manipulating the missingness process to induce bias optimally is computationally hard. However, a simple heuristic can effectively introduce bias to an independent imputation model that simply tries to maximize imputation accuracy. We formulate the problem for a given number of observed sensitive values $k$.

**Definition 1.** *Adversarial Missingness Against Fair Classification (AMAFC): Given a graph $\mathcal{G} = (\mathcal{V}, \mathcal{E})$, select a set of nodes $\mathcal{V}^*_S$ that maximizes the bias of a fair classifier with parameters $\theta^*_{class}$ trained with an imputation model with parameters $\theta^*_{imp}$ trained with $\mathcal{V}_S$:*

$$\mathcal{V}^*_S = \arg\max_{\mathcal{V}_S \in \mathcal{V}, |\mathcal{V}_S| = k} \mathcal{L}_{bias}(\theta^*_{class}, \mathcal{V})$$
$$s.t. \quad \theta^*_{class} = \arg\min_{\theta_{class}} \mathcal{L}_{class}(\theta_{class}) + \alpha\mathcal{L}_{bias}(\theta_{class}, \theta^*_{imp}, \mathcal{V}_S)$$
$$s.t. \quad \theta^*_{imp} = \arg\min_{\theta_{imp}} \mathcal{L}_{imp}(\theta_{imp}, \mathcal{V}_S)$$

where we assume that the adversary can compute the bias $\mathcal{L}_{bias}(\theta^*_{class}, \mathcal{V})$ based on all sensitive attributes $s_v$, while the classifier can only estimate its bias based on a combination of nodes $\mathcal{V}_S$ with observed $s_v$ and imputed values produced with $\theta^*_{imp}$. While AMAFC describes the objective of an idealized adversary, it is impractical due to its associated complexity (tri-level optimization).

**Definition 2.** *Adversarial Missingness against Data Bias (AMADB): Given a graph $\mathcal{G} = (\mathcal{V}, \mathcal{E})$, select a set of nodes $\mathcal{V}^*_S$ that minimizes the bias in the labels estimated using an imputation model with parameters $\theta^*_{imp}$ trained with $\mathcal{V}_S$:*

$$\mathcal{V}^*_S = \arg\min_{\mathcal{V}_S \in \mathcal{V}, |\mathcal{V}_S| = k} \mathcal{L}_{bias}(\theta^*_{imp}, \mathcal{V}_S, \mathcal{V}_L)$$
$$s.t. \quad \theta^*_{imp} = \arg\min_{\theta_{imp}} \mathcal{L}_{imp}(\theta_{imp}, \mathcal{V}_S)$$

where $\mathcal{L}_{bias}(\theta^*_{imp}, \mathcal{V}_S, \mathcal{V}_L)$ is computed based on labels instead of $f_{class}$. By minimizing the bias computed based on imputed values and labels, AMADB attempts to misguide any classifier that relies on such imputation to mitigate bias. AMADB is more tractable than AMFC, but still NP-hard.

**Theorem 1.** *The AMADB problem is NP-hard.*

See the proof in the Appendix. We frame AMADB as an adversarial version of *active learning* where the goal is to strategically minimize the accuracy of the imputation by selecting observed sensitive attributes. More specifically, we apply a popular formulation of active learning as a coverage problem (Yehuda et al., 2022; Ren et al., 2021). Theorem 1 can be interpreted as a positive result, as it shows that, in theory, a simpler surrogate of the adversary's objective is still hard to optimize.

**A simple (yet effective) heuristic for adversarial missingness:** Let $s \in \{0,1\}$ denote the true sensitive attribute and $\hat{s} \in \{0,1\}$ be its imputation. We define the imputation error rate as $p(s \neq \hat{s}) = \epsilon$. Let $\hat{y} \in \{0,1\}$ be the prediction of a fair model that minimizes demographic parity ($\Delta DP$) with respect to $\hat{s}$, where we define $\Delta DP$ with respect to a sensitive attribute $a \in \{s, \hat{s}\}$ as follows:

$$\Delta DP_a = |p(\hat{y} = 1|a = 0) - p(\hat{y} = 1|a = 1)|$$

We want to design a heuristic where the goal of the adversary is to choose missing values to maximize the difference between the true demographic parity $\Delta DP_s$ and empirical demographic parity $\Delta DP_{\hat{s}}$:

$$\max_{\hat{s}} \ \Delta DP_s - \Delta DP_{\hat{s}}$$
$$= \max_{\hat{s}} \ |p(\hat{y} = 1|s = 0) - p(\hat{y} = 1|s = 1)| - |p(\hat{y} = 1|\hat{s} = 0) - p(\hat{y} = 1|\hat{s} = 1)|$$

We assume the minority class is underrepresented, i.e. $(\forall a, \ |p(\hat{y} = 1|a = 0) \geq p(\hat{y} = 1|a = 1)|)$:

$$\max_{\hat{s}} \ p(\hat{y} = 1|s = 0) - p(\hat{y} = 1|s = 1) - p(\hat{y} = 1|\hat{s} = 0) + p(\hat{y} = 1|\hat{s} = 1)$$
$$= \max_{\hat{s}} \ p(\hat{y} = 1|s = 0) - p(\hat{y} = 1|\hat{s} = 0) + p(\hat{y} = 1|\hat{s} = 1) - p(\hat{y} = 1|s = 1)$$
$$= \max_{\hat{s}} \ p(\hat{y} = 1|s = 0) - p(\hat{y} = 1|\hat{s} = 0) + p(\hat{y} = 0|s = 1) - p(\hat{y} = 0|\hat{s} = 1)$$

Assuming $p(\hat{y} \neq y) \rightarrow 0$, the adversary should attack nodes with $y = 1 \wedge s = 0$ and $y = 0 \wedge s = 1$ to maximize $p(s \neq \hat{s}|y = 1, s = 0)$ and $p(s \neq \hat{s}|y = 0, s = 1)$. Based on Theorem 1, this problem is NP-hard. It also requires the adversary to have access to true class labels and sensitive values, which is a strong assumption. For practicality, we design an efficient adversary that aims to increase the imputation error $\epsilon = p(s \neq \hat{s})$ by exploiting the degree bias of GNNs using only the graph topology.

**Definition 3.** *The degree bias assumption: Given nodes $u, v \in \mathcal{V}$ where $deg(u) > deg(v)$, we assume that $p(s_v \neq \hat{s}_v) > p(s_u \neq \hat{s}_u)$.*

The degree bias assumption has been supported by both theoretical and empirical results in the literature for both homophilic and non-homophilic graphs (Tang et al., 2020; Liu et al., 2023; Ju et al., 2024; Subramonian et al., 2024). Moreover, low-degree nodes are known to be more vulnerable to attacks than high-degree nodes (Zügner et al., 2018; Zügner & Günnemann, 2019). Therefore, our adversarial missingness process simply selects low-degree nodes to have missing sensitive values. We call the heuristic based on the degree bias assumption the 'degree' heuristic.

We also consider another heuristic to evaluate the effectiveness of the degree one. We select missing values uniformly at random from the set $\mathcal{V}_{miss} = \{v \in \mathcal{V}|(y_v = 1 \wedge s_v = 0) \vee (y_v = 0 \wedge s_v = 1)\}$. If the desired number of missing nodes exceeds $|\mathcal{V}_{miss}|$, we draw the remaining $|\mathcal{V} \setminus \mathcal{V}_S| - |\mathcal{V}_{miss}|$ samples uniformly at random from $\mathcal{V} \setminus \mathcal{V}_{miss}$. We call the resulting heuristic 'targeted'.

To assess the effect of degree-based adversarial missingness, we compare imputation under adversarial degree, targeted and random missingness. Figure 2 reports correlations between sensitive attributes and class labels across datasets. Using a G Convolutional Network (GCN) (Kipf & Welling, 2017) for independent imputation, we observe that with less than 50% sensitive data, adversarial missingness consistently underestimates the true bias in the dataset, while random missingness and targeted missingness perform slightly better (Figures 2, top row). In contrast, BFtS (Figures 2, bottom row), introduced in the next section, rarely underestimates bias. These results highlight that off-the-shelf imputation can mislead fair graph learning by underestimating bias in the dataset, particularly under adversarial missingness. Among the three missingness processes, the degree heuristic exhibits the largest bias discrepancy between imputed and original values (i.e., is more adversarial). Therefore, we focus our analysis on the degree heuristic.

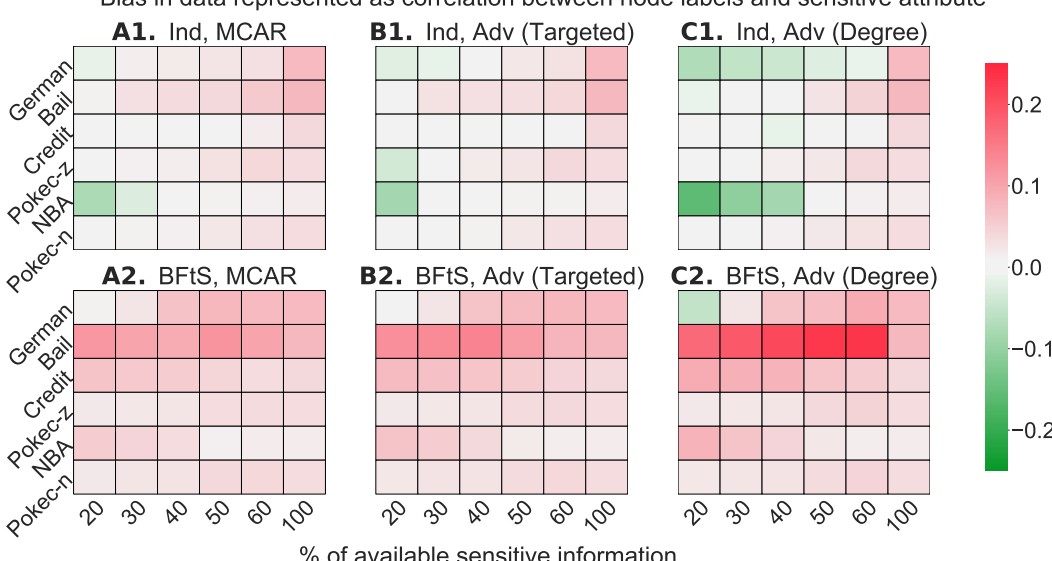

Figure 2: Empirical evaluation of three missingness processes and their effect on bias estimation. Degree-based adversarial missingness most effectively drives an independently trained GCN imputation model to underestimate bias, compared with random and targeted missingness. The final column in each matrix reports the true bias, where lower values indicate stronger underestimation. NBA shows the largest correlation gap (up to a 433% discrepancy when only 20% of protected attributes are observed) under the degree heuristic. We also include results for BFtS (Section 4), our 3-player adversarial imputation framework.

## 4 FAIRNESS-AWARE ADVERSARIAL MISSING DATA IMPUTATION

We introduce BFtS (Better Fair than Sorry), a 3-player adversarial framework for fair GNN training under adversarially missing sensitive data. The fair GNN is trained jointly with two adversaries–one to predict sensitive attributes based on GNN embeddings and another to impute missing values that minimize fairness, ensuring that fairness is evaluated against worst-case imputations.

We motivate the worst-case assumption using distributionally robust optimization (Ben-Tal et al., 2013; Mandal et al., 2020; Namkoong & Duchi, 2016; Shafieezadeh Abadeh et al., 2015). Let $\mathcal{P}_s$ be the sensitive attribute distribution. We can express the fairness objective in terms of the expected bias.

$$\theta^*_{class} = \arg \min_{\theta_{class}} \mathcal{L}_{class} + \alpha \mathbb{E}_{s \sim \mathcal{P}_s}[\mathcal{L}_{bias}]$$

However, under adversarial missingness, the true distribution of sensitive values cannot be accurately estimated from the observations (Mohan et al., 2013; Tian, 2017). To handle this uncertainty, let us define an uncertainty set $\mathcal{U}$ of plausible distributions for $s$. This leads to the worst-case imputation:

$$\theta^*_{class} = \arg \min_{\theta_{class}} \mathcal{L}_{class} + \alpha \max_{u \in \mathcal{U}} \mathbb{E}_{s \sim u}[\mathcal{L}_{bias}]$$

### 4.1 PROPOSED MODEL (BFTS)

Our goal is to learn a group fair and accurate GNN $\hat{y} = f_{class}(\mathcal{G}, \mathcal{X})$ for node classification with adversarially missing sensitive values. The proposed solution is model-agnostic (e.g., GNN) and based on adversarial learning. It formulates the imputation model as a second adversary of the GNN. Figure 3 depicts the flow diagram of the proposed model. The model has three primary components: a missing sensitive attribute imputation GNN $f_{imp}$, a node classification GNN $f_{class}$, and a sensitive attribute Deep Neural Net (DNN) prediction $f_{bias}$. $f_{class}$ takes $\mathcal{X}$ and $\mathcal{G}$ as inputs and predicts the node labels. $f_{bias}$ is an adversarial neural network that attempts to estimate the sensitive information from the final layer representations of $f_{class}$ to assess the bias. More specifically, $f_{class}$ is biased if

the adversary $f_{bias}$ can accurately predict the sensitive attribute information from the representations of $f_{class}$. The model $f_{imp}$ predicts the missing sensitive attributes by taking $\mathcal{X}$ and $\mathcal{G}$ as inputs and generating the missing sensitive attributes $\hat{si}$. The goal of $f_{imp}$ is to generate sensitive values that minimize the fairness of $f_{class}$, and, thus, it works as a second adversary to $f_{class}$.

### 4.1.1 PLAYER ARCHITECTURES

**GNN classifier $f_{class}$:** Node classification model $\hat{y}_v = f_{class}(x_v, \mathcal{G})$ implemented using a GNN. We assume that $f_{class}$ does not apply sensitive attributes but uses other attributes in $\mathcal{X}$ that might be correlated with sensitive ones. The goal of $f_{class}$ is to achieve both accuracy and fairness. To improve fairness, $f_{class}$ tries to minimize the loss of adversary $f_{bias}$.

**Sensitive attribute predictor (Adversary 1) $f_{bias}$:** Neural network that uses representations $\mathbf{h}_v$ from $f_{class}$ to predict sensitive attributes as $\hat{sa}_v = f_{bias}(\mathbf{h}_v)$. $f_{class}$ is fair if $f_{bias}$ performs poorly.

**Missing data imputation GNN (Adversary 2) $f_{imp}$:** Predicts missing sensitive attributes as $\hat{si}_v = f_{imp}(x_v, \mathcal{G})$. However, besides being accurate, $f_{imp}$ plays the role of an adversary to $f_{class}$ by predicting values that maximize the accuracy of $f_{bias}$.

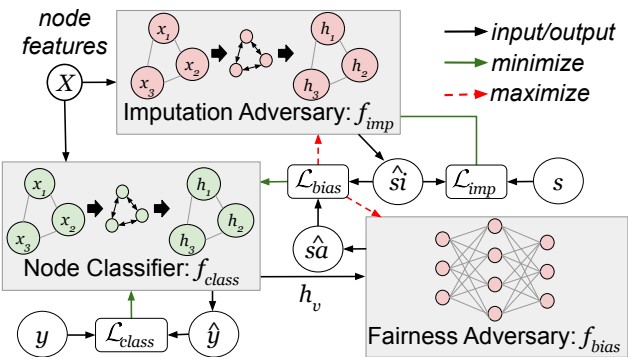

Figure 3: 3-player framework for fair GNN training with missing data imputation (BFtS). $f_{class}$ generates node representations $\mathbf{h}_v$ by minimizing the classification loss $\mathcal{L}_{class}$ (Eqn. 1) and the maximizing sensitive attribute prediction loss $\mathcal{L}_{bias}$ (Eqn. 3). $f_{bias}$ predicts sensitive attributes using representations from $f_{class}$ by minimizing $\mathcal{L}_{bias}$. $f_{imp}$ predicts missing values by minimizing the imputation loss $\mathcal{L}_{imp}$ (Eqn. 2) and maximizing $\mathcal{L}_{bias}$. $\hat{y}$, $\hat{si}$, $\hat{sa}$ are predictions from $f_{class}$, $f_{imp}$ and $f_{bias}$, respectively.

### 4.1.2 LOSS FUNCTIONS

**Node Classification:** We apply the cross-entropy loss to learn $f_{class}$ as follows:

$$\mathcal{L}_{class} = -\frac{1}{|\mathcal{V}_L|} \sum_{v \in \mathcal{V}_L} y_v log(\hat{y}_v) + (1 - y_v)log(1 - \hat{y}_v). \tag{1}$$

**Sensitive Attribute Imputation:** Because sensitive attributes tend to be imbalanced, $f_{imp}$ applies the Label-Distribution-Aware Margin (LDAM) loss Cao et al. (2019). Let $\hat{si} = f_{imp}(X, G)$ be the one-hot encoded predictions and $\hat{si}_v^s$ be the prediction for $s_v$. The LDAM loss of $f_{imp}$ is defined as:

$$\mathcal{L}_{imp} = \frac{1}{|\mathcal{V}_S|} \sum_{v=1}^{|\mathcal{V}_S|} -log\frac{e^{\hat{si}_v^s - \Delta^s}}{e^{\hat{si}_v^s - \Delta^s} + \sum_{k \neq s} e^{\hat{si}_v^k}}. \tag{2}$$

where $\Delta^j = C/n_j^{\frac{1}{4}}$ and $j \in \{0, 1\}$, $C$ is a constant independent of the sensitive attribute $s$, and $n_j$ is the number of samples that belong to $s = j$.

The LDAM loss is a weighted version of the negative log-likelihood loss. Intuitively, since $\Delta^j$ is larger for smaller values of $n_j$, and thus it ensures a higher margin for the smaller classes.

**Sensitive attribute imputation with no sensitive information:** BFtS can generate fair outputs when very little or no sensitive information is provided by letting $f_{imp}$ impute the sensitive values from the ground truth training labels $y$ using the LDAM loss Du et al. (2021). More specifically, we replace $\mathcal{V}_S$ with $\mathcal{V}_L$ and $s$ with $y$ for each node $v \in \mathcal{V}_L$ in Eqn 2. The reasoning is that the worst-case fairness model generally assigns a more desired outcome to the non-sensitive group and a less desired outcome to the sensitive one. Therefore, the nodes predicted as the minority class will fall into the sensitive group, and vice versa. This is consistent with the worst-case assumption of BFtS.

**Sensitive Attribute Prediction:** Given the sensitive information for $\mathcal{V}_S$, we first replace $\hat{si}_v$ by $s_v$ for $v \in \mathcal{V}_S$ and thereby generate $\hat{s}$. The parameters of $f_{bias}$ are learned using:

$$\mathcal{L}_{bias} = \mathbb{E}_{\mathbf{h} \sim p(\mathbf{h}|\hat{s}=1)}[\log f_{bias}(\mathbf{h})] + \mathbb{E}_{\mathbf{h} \sim p(\mathbf{h}|\hat{s}=0)}[\log(1 - f_{bias}(\mathbf{h}))] \tag{3}$$

For simplicity, we consider a binary sensitive attribute in this paper. However, BFtS extends to non-binary or continuous sensitive attributes by appropriately generalizing the imputation loss $\mathcal{L}_{imp}$ and the bias loss $\mathcal{L}_{bias}$ to multi-class or continuous settings.

### 4.1.3 LEARNING THE PARAMETERS OF BFTS

Let $\theta_{class}, \theta_{bias}$, and $\theta_{imp}$ be the parameters of $f_{class}$, $f_{bias}$, and $f_{imp}$, respectively, which are learned via a 3-player adversarial scheme described in Figure 3. Parameters $\theta_{class}$ are optimized as:

$$\theta_{class}^* = \arg \min_{\theta_{class}} \mathcal{L}_{class} + \alpha \mathcal{L}_{bias}. \tag{4}$$

where $\alpha$ is a hyperparameter that controls the trade-off between accuracy and fairness.

The parameters of the sensitive attribute predictor $\theta_{bias}$ are learned by maximizing $\mathcal{L}_{bias}$:

$$\theta_{bias}^* = \arg \max_{\theta_{bias}} \mathcal{L}_{bias}. \tag{5}$$

To learn the parameters $\theta_{imp}$ of the imputation model to generate predictions that are accurate and represent the worst-case scenario for fairness, we apply the following:

$$\theta_{imp}^* = \arg \min_{\theta_{imp}} \mathcal{L}_{imp} - \beta \mathcal{L}_{bias}. \tag{6}$$

Here $\beta$ is a hyperparameter that controls the trade-off between imputation accuracy and worst-case imputation. The min-max objective between the three players is therefore:

$$\min_{\theta_{class}} \max_{\theta_{imp}, \theta_{bias}} \mathbb{E}_{\mathbf{h} \sim p(\mathbf{h}|\hat{s}=1)}[\log f_{bias}(\mathbf{h})] + \mathbb{E}_{\mathbf{h} \sim p(\mathbf{h}|\hat{s}=0)}[\log(1 - f_{bias}(\mathbf{h}))]. \tag{7}$$

The training algorithm and time complexity analysis of BFtS are discussed in Appendix.

### 4.2 THEORETICAL ANALYSIS

We will analyze some theoretical properties of BFtS. All the proofs are provided in Appendix.

**Theorem 2.** *BFtS learns an imputation model $f_{imp}$ with the worst-case imputation:*

$$\theta_{imp}^* = \arg \max_{\theta_{imp}} \mathcal{L}_{bias} = \arg \max_{\theta_{imp}} |p(\hat{y} = 1|\hat{s} = 1) - p(\hat{y} = 1|\hat{s} = 0)|$$

where $\hat{y} = f_{class}(\mathcal{G}, \mathcal{X})$ and, $\hat{s} = f_{imp}(\mathcal{G}, \mathcal{X})$

The theorem shows that $f_{imp}$ indeed generates imputations with minimum fairness (or maximum bias) based on Demographic Parity for a given classifier $f_{class}$.

**Theorem 3.** *BFtS learns a classifier $f_{class}$ that minimizes the worst-case bias:*

$$\theta_{class}^* = \arg \min_{\theta_{class}} \sup_{\theta_{imp}} |p(\hat{y} = 1|\hat{s} = 1) - p(\hat{y} = 1|\hat{s} = 0)|$$

The optimal GNN classifier $f_{class}$, will achieve demographic parity ($\Delta DP = 0$ in Sec 5) for the worst-case imputation (minimum fairness) $\hat{s}$ generated by $f_{imp}$. As $f_{class}$ is a minimax estimator, the maximal $\Delta DP$ of BFtS is minimum amongst all estimators of $s$.

**Corollary 1.** *Let $s' = f(\mathcal{G}, \mathcal{X})$ be an imputation method independent of models $f_{class}$ and $f_{bias}$, then the BFtS imputation $\hat{s} = f_{imp}(\mathcal{G}, \mathcal{X})$ is such that:*

$$JS(p(\mathbf{h}|s' = 1); p(\mathbf{h}|s' = 0)) \leq JS(p(\mathbf{h}|\hat{s} = 1); p(\mathbf{h}|\hat{s} = 0))$$

where $JS$ is the *Jensen Shannon divergence*.

The value of $JS(p(\mathbf{h}|s'=1); p(\mathbf{h}|s'=0))$ is related to the convergence of adversarial learning. For independent imputation, if $s'$ is inaccurate, then $JS(p(\mathbf{h}|s'=1); p(\mathbf{h}|s'=0)) \approx 0$ and $\mathcal{L}_{bias}$ in Eq. 4 will be constant. The interplay between the three players makes BFtS more robust to convergence issues because the objective minimizes the upper bound on the JS divergence. This reduces the probability that the divergence vanishes during training (see details in the Appendix).

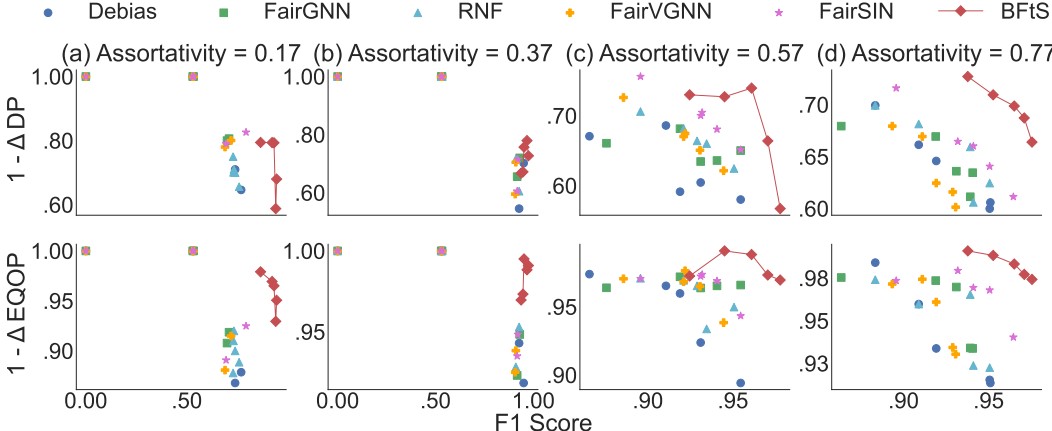

Figure 4: Performance of the methods using the SIMULATION dataset for different values of assortativity coefficients. In the x-axis, we plot the F1 score, and in the y-axis of the top row, we plot $1 - \Delta DP$, and in the y-axis of the bottom row, we plot $1 - \Delta EQOP$. The top right corner of the plot, therefore, represents a high F1 with low bias. When the assortativity is low, other methods fail to learn the node labels. With higher assortativity, though other methods learn the class labels, BFtS is less biased and has similar accuracy. Note that the X-axes have different ranges.

## 5 EXPERIMENTAL EVALUATION

We compare our approach (BFtS) against alternatives in terms of accuracy and fairness using real and synthetic data. We apply average precision (AVPR) and F1 for accuracy evaluation and $\Delta DP$ and $\Delta EQOP$ for fairness. Details about datasets, baselines, evaluation, and hyperparameters are provided in the Appendix. We group the baselines into two categories. First, we compare with methods that are explicitly designed to handle missing values: Debias Buyl & De Bie (2020), FairGNN Dai & Wang (2021), and RNF Du et al. (2021). Among these, RNF can operate when all sensitive attributes are missing, whereas Debias and FairGNN require access to at least some sensitive information. Both RNF and FairGNN rely on independent sensitive-attribute imputation procedures. Second, we compare our approach with state-of-the-art fair GNN models, including FairSIN Yang et al. (2024) and FairVGNN Wang et al. (2022). These methods cannot accommodate missing values directly, so we apply an independent imputation strategy consistent with the procedure used in FairGNN to supply the missing sensitive attributes before training. Additional experiments varying the GNN, scalability and complexity analysis using a synthetic large-scale graph, ablation studies (hyperparameter sensitivity, impact of LDAM loss, and impact of varying $\mathcal{V}_S$), and visualization of learnt representation and worst case assumption trade-off are also included in the Appendix. Our code can be found here: `https://github.com/DebolinaHalder/BFtS`.

### 5.1 RESULTS AND ANALYSIS

Figure 4 shows the F1 score, $1 - \Delta DP$, and $1 - \Delta EQOP$ for different methods while varying their hyperparameters. The SIMULATION graph was generated using a stochastic block model with different assortativity coefficients, i.e., the extent to which links exist within clusters compared with across clusters. Learning missing sensitive values and node labels under low assortativity is hard, and graphs with assortativity 0.17 and 0.37 represent the scenario described in Corollary 1, therefore, the JS divergence tends to be small for independent imputation, which may result in a lack of convergence

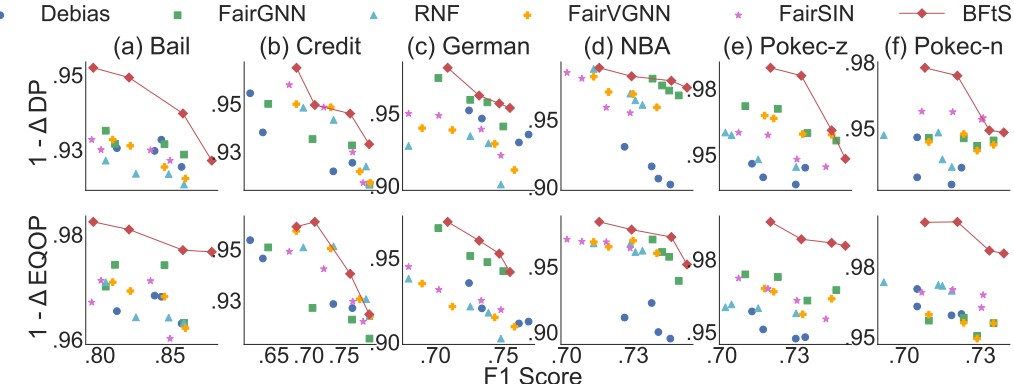

Figure 5: Fairness vs. accuracy results. The x-axis of each plot shows the F1 score. We plot $1 - \Delta DP$ and $1 - \Delta EQOP$ on the y-axis of rows 1 and 2, respectively. The top right corner of the plot represents a high F1 with low bias. BFtS often achieves better fairness for a similar value of F1.

| | RNF | | | | BFtS | | | |
|---|---|---|---|---|---|---|---|---|
| | %AVPR(↑) | %F1(↑) | %ΔDP(↓) | %ΔEQ(↓) | %AVPR(↑) | % F1(↑) | %ΔDP (↓) | %ΔEQ(↓) |
| BAIL | 81.0±0.1 | 85.3±0.3 | 11.65±0.02 | 8.51±0.02 | **83.1±0.2** | **86.2±0.3** | **8.01±0.03** | **4.12±0.01** |
| CREDIT | 80.4±0.4 | 75.9±0.1 | 8.01±0.08 | 7.25±0.08 | **82.1±0.4** | **76.8±0.0** | **5.97±0.10** | **4.96±0.06** |
| GERMAN | 73.2±0.2 | 74.4±0.1 | 9.42±0.04 | 8.92±0.12 | **74.1±0.3** | **74.7±0.4** | **6.42±0.11** | 7.68±0.17 |
| NBA | 70.1±0.2 | **70.2±0.1** | 6.47±0.03 | 5.89±0.05 | **72.9±0.3** | 69.8±0.1 | **5.19±0.01** | **3.59±0.05** |
| POKEK-Z | 73.1±0.4 | 71.2±0.1 | 6.18±0.05 | 6.29±0.07 | **73.4±0.2** | **73.2±0.3** | **5.10±0.02** | **3.20±0.06** |
| POKEK-N | 71.6±0.2 | **72.2±0.1** | 7.58±0.01 | 7.09±0.04 | **72.6±0.1** | 69.6±0.4 | **4.29±0.02** | **3.01±0.03** |

Table 1: AVPR, F1, $\%\Delta DP$, and $\%\Delta EQOP$ without any sensitive information for BFtS and RNF (only baseline that operates in this setting). BFtS outperforms RNF in terms of fairness and accuracy.

for the baselines (FairGNN and Debias). As we increase the assortativity, all baselines can predict the labels, but BFtS still achieves a better fairness vs. accuracy trade-off in all cases.

We demonstrate the accuracy vs. fairness trade-off for real datasets in Figure 5. The top right corner of the plot represents high fairness and accuracy. Our model achieves better fairness and similar accuracy to the best baseline for the BAIL, NBA, GERMAN, POKEC-N and POKEC-Z. For all datasets, BFtS achieves a better fairness-accuracy tradeoff.

Table 1 shows the accuracy and fairness results for BAIL, CREDIT, GERMAN, POKEC-Z, POKEK-N and NBA without any sensitive attribute information. Among the baselines, only RNF works in this setting. BFtS outperforms RNF for the BAIL, GERMAN, and CREDIT. BFtS also outperforms RNF in terms of fairness on POKEC-Z, POKEC-N, and NBA with similar AVPR and F1.

BFtS outperforms the baselines because it effectively implements the worst-case assumption for missing value imputation. This assumption leads to stronger fairness guarantees (Theorem 3) than the baselines in cases where missing values are adversarial.

# 6 CONCLUSION

We investigate the challenge of incorporating fairness considerations into graph machine learning models when sensitive attributes are missing due to adversarial processes. Our solution is BFtS, a 3-player adversarial learning framework for the imputation of adversarially missing sensitive attributes that produce challenging values for graph-based fairness. Theoretical and empirical results demonstrate that BFtS achieves a better fairness $\times$ accuracy trade-off than existing alternatives.

As future work, we want to investigate fair graph machine learning that incorporates fairness considerations beyond the ones considered here. In particular, we are interested in metrics that are more sensitive to the ranking of predictions Mattos et al. (2026); Han et al. (2025).

ACKNOWLEDGEMENTS

We acknowledge the support by the US Department of Transportation Tier-1 University Transportation Center (UTC) Transportation Cybersecurity Center for Advanced Research and Education (CYBER-CARE) (Grant No. 69A3552348332), and the Rice Ken Kennedy Institute.

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

## A    TECHNICAL APPENDICES AND SUPPLEMENTARY MATERIAL

This supplementary material includes the following:

- Details of Graph Neural Network (GNN)
- Evaluation Metrics
- Dataset Details
- Baselines
- Software and Hardware
- Hyperparameter Setting
- Proof of theorems and corollaries
- Training algorithm
- Complexity and Running time comparison
- Imputation accuracy
- Worst case fairness accuracy trade-off
- Additional experiments
- Ablation Study

## GRAPH NEURAL NETWORK

Two of the models (or players) learned by our model are GNNs. Without loss of generality, we will assume that they are message-passing neural networks (Gilmer et al., 2017), which we will simply refer to as GNNs. A GNN learns node representations $\mathbf{h}_v$ for each node $v$ that can be used to predict (categorical or continuous) values $y_v = f(\mathbf{h}_v)$. These representations are learned via successive AGGREGATE and COMBINE operations defined as follows:

$$\mathbf{h}_v^k = \mathsf{COMBINE}^k(\mathbf{h}_v^{k-1}, \mathsf{AGGREGATE}^{k-1}(\mathbf{h}_u^{k-1}, A_{u,v}=1)).$$

where $\mathbf{h}_v^k$ are representations learned at the $k$-th layer.

## EVALUATION METRICS

**Utility Evaluation:** As with many datasets in the fairness literature, our datasets are imbalanced. To evaluate their accuracy, we apply average precision (AVPR) and the F1 score.

**Fairness Evaluation:** We apply two bias metrics that will be defined based on a sensitive attribute $s$. Let $s = 1$ be the sensitive group and $s = 0$ be the non-sensitive group. The *Demographic Parity (DP)* requires the prediction to be independent of the sensitive attribute (Kamiran & Calders, 2009). We evaluate demographic parity as:

$$\Delta DP(\hat{y}, s) = |p(\hat{y} = 1|s = 1) - p(\hat{y} = 1|s = 0)|. \tag{8}$$

*Equality of Opportunity (EQOP)* calculates the difference between true positive rates between sensitive and non-sensitive groups (Hardt et al., 2016) as follows:

$$\Delta EQOP(\hat{y}, s, y) = |p(\hat{y} = 1|s = 1, y = 1) - p(\hat{y} = 1|s = 0, y = 1)|. \tag{9}$$

*Equalized Odds (EQODDs)* calculates the average difference between true positive rates and false positive rates between sensitive and non-sensitive groups (Grant, 2023) as follows:

$$\Delta EQODDs(\hat{y}, s, y) = \frac{1}{2}|p(\hat{y} = 1|s = 1, y = 1) - p(\hat{y} = 1|s = 0, y = 1)| + \\ |p(\hat{y} = 1|s = 1, y = 0) - p(\hat{y} = 1|s = 0, y = 0)|. \tag{10}$$

We can get corresponding fairness by subtracting $\Delta DP$ and $\Delta EQOP$ from 1.

We use $\Delta DP$ and $\Delta EQOP$ for most of our analysis as they are commonly used group-based metrics in fairness literature. Our theoretical analysis guarantees Demographic Parity, but we can change the $\mathcal{L}\_bias$ loss to account for a different type of fairness Madras et al. (2018).

DATASET DETAILS

SIMULATION: We simulate a synthetic graph consisting of 1000 nodes based on the stochastic block model with blocks of sizes 600 and 400, for the majority and minority classes, respectively. We can have different edge probabilities within and across blocks and generate graphs with different assortativity coefficients. The sensitive attribute $s$ is simulated as $s \sim Bernoulli(p)$ if $y = 1$ and $s \sim Bernoulli(1 - p)$ otherwise. The probability $p = 0.5$ generates a random assignment of the sensitive attribute, while $p$ close to 1 or 0 indicates an extremely biased scenario. For our simulation, we use $p = 0.7$ so that the dataset is unfair. Each node has 20 attributes, of which 8 are considered noisy. The noisy attributes are simulated as $Normal(0, 1)$ and the remaining ones as $\gamma * y + Normal(0, 1)$, where $\gamma$ is the strength of the signal in important features.

BAIL (Jordan & Freiburger, 2015): Contains 18,876 nodes representing people who committed a criminal offense and were granted bail by US state courts between 1990 and 2009. Defendants are linked based on similarities in their demographics and criminal histories. The objective is to categorize defendants as likely to post bail vs. no bail. No-bail defendants are expected to be more likely to commit violent crimes if released. Race information is considered a sensitive attribute. Features include education, marital status, and profession.

CREDIT (Yeh & Lien, 2009): The dataset has $30,000$ nodes representing people each with $25$ features, including education, total overdue counts, and most recent payment amount. The objective is to forecast if a person will or won't miss a credit card payment while considering age as a sensitive attribute. Nodes are connected based on payment patterns and spending similarities.

GERMAN (Hofmann, 1994): The dataset consists of 1000 nodes and 20 features (e.g., credit amount, savings) representing clients at a German bank. Labels are credit decisions for a person (good or bad credit). The sensitive attribute is gender. The goal is to predict who should receive credit.

NBA (Dai & Wang, 2021): Dataset based on a Kaggle dataset about the National Basketball Association (US). It contains 403 players as nodes connected based on their Twitter activity. Attributes represent player statistics and nationality is considered the sensitive attribute. The goal is to predict the players' salaries (above/under the median).

POKEC (Takac & Zabovsky, 2012): This dataset has two variations: POKEC-Z and POKEC-N. The pokec dataset is based on a popular social network in Slovakia. The sensitive attribute in POKEC-Z and POKEC-N are two different regions of Slovakia. The goal is to predict the working field of users.

BASELINES

Vanilla: The vanilla model is the basic GNN model trained with cross-entropy loss without any fairness intervention.

Debias (Zhang et al., 2018): Debias is an adversarial learning method that trains a classifier and an adversary simultaneously. The adversary is trained using the softmax output of the last layer of the classifier. It does not handle missing sensitive data. Hence, we only use $\mathcal{V}_S \in \mathcal{V}$ to calculate the adversarial loss.

FairGNN (Dai & Wang, 2021): FairGNN first uses a GNN to estimate the missing sensitive attributes. It then trains another GNN-based classifier and a DNN adversary. The adversary helps to eliminate bias from the representations learned by the GNN classifier.

RNF (Du et al., 2021): RNF imputes the missing sensitive attributes by first training the model with generalized cross entropy (Zhang & Sabuncu, 2018). It eliminates bias from the classification head. RNF uses samples with the same ground-truth label but distinct sensitive attributes and trains the classification head using their neutralized representations.

FairVGNN (Wang et al., 2022): FairVGNN learns fair node representations via automatically identifying and masking sensitive-correlated features. It requires complete sensitive information, so we use the same imputation method as FairGNN for the missing values.

FairSIN (Yang et al., 2024): FairSIN learns fair representations by adding additional fairness-facilitating features. It also requires complete sensitive information, so we use the same independent imputation method as FairGNN for the missing values. (Yang et al., 2024):

## SOFTWARE AND HARDWARE

- Operating System: Linux (Red Hat Enterprise Linux 8.9 (Ootpa))
- GPU: NVIDIA A40
- Software: Python 3.8.10, torch 2.2.1, dgl==0.4.3

## HYPERPARAMETER SETTINGS

The size of $\mathcal{V}_S$ was set to $30\%$ of $|\mathcal{V}|$ unless stated otherwise. Each GNN has two layers and a dropout probability of $0.5$. In practice, some sensitive information is often available, unless otherwise stated. Therefore, we concentrate the majority of our experiments on the assumption that there is a limited amount of sensitive information available, and we generate the imputations using the available sensitive information in Equation 2. The $\mathcal{V}_S$ was set to $30\%$ for all experiments unless stated otherwise. $\mathcal{V}_L$ is $300, 400, 1400, 100, 500, 500$, and $800$ for the SIMULATION, GERMAN, CREDIT, NBA, POKEC-Z, POKEC-N and BAIL datasets, respectively. We adjust the values of $\alpha$ and $\beta$ for each dataset using cross-validation based on the F1 score. For all baselines, we choose the respective regularizers using cross-validation based on the same metrics. Every model, except RNF, combines the training of several networks with various learning rates. We stop training RNF based on early stopping to eliminate overfitting. We train the rest of the models for certain epochs and choose the model with the best AVPR. All the results shown in the experiment section are the average of 5 runs. We use GCN as the GNN architecture for all experiments unless mentioned otherwise.

## PROOF OF THEOREMS

### THEOREM 1

We will consider an adversarial active learning setting, where the goal of the adversary is to select $k$ labels (i.e., observed sensitive values) to minimize the bias $\mathcal{L}_{bias}$ of the data imputed by $f_{imp}$. More specifically, our proof will use the coverage model for active learning, where an unlabeled data point $x_i^u$ can be correctly predicted iff it is covered by at least one labeled point $x_j^l$. The goal is for the labeled points $\{x_1^l, x_2^l, \ldots x_k^l\}$ to cover as many unlabeled data points $\{x_1^u, x_2^u, \ldots x_{n-k}^u\}$ as possible (Yehuda et al., 2022; Ren et al., 2021). Thus, in our setting, the adversary's goal is to select labels to minimize the coverage of the following sets of nodes: (1) nodes in the sensitive group ($s_v = 1$) and the negative class ($y = 0$) and (2) nodes in the non-sensitive group ($s_v = 0$) and the positive class ($y = 1$). This can be achieved by minimizing the coverage of such nodes. We provide a reduction from the *minimum k-union problem*.

**Minimum k-union problem (MkU, Chlamtáč et al. (2017)):** Given a set of sets $\mathcal{S} = S_1, S_2, \ldots S_q$, where each set $S_i \subseteq \mathcal{I}$ and $\mathcal{I}$ is a set of items. The problem consists of selecting $r$ sets $S_1, S_2, \ldots S_r$ from $\mathcal{S}$ to minimize the coverage $S_1 \cap S_2 \cap \ldots S_r$.

Under our coverage model, the reduction is straightforward. Given an instance of *MkU*, we define a corresponding instance of the *Adversarial Missingness against Data Bias (AMADB)* problem as follows. We represent each item in $\mathcal{I}$ and each set in $\mathcal{S}$ as a data point $x$ and assign all data points to both the non-sensitive group ($s_v = 0$) and the positive class ($y = 1$). Moreover, we will add one data point $x'$ that will belong to the sensitive group ($s_v = 1$) and the positive class ($y = 1$). Therefore, the resulting set of data points is $\{x_1, x_2, \ldots x_n\}$, where $n = |\mathcal{S}| + |\mathcal{I}| + 1$. Moreover, let $x_i$ cover $x_j$ iff $x_i$ represents a set $S_i \in \mathcal{S}$, $x_j$ represents an item $i_j \in \mathcal{I}$, and $i_j \in S_i$. It follows that minimizing the bias in the AMADB instance by selecting data points $x_j$ to be labeled is equivalent to minimizing the coverage in the corresponding *MkU* instance.

### THEOREM 2

*Proof.* From Eq. 7, the min-max objective of BFtS is:

$$\mathbb{E}_{\mathbf{h} \sim p(\mathbf{h}|\hat{s}=1)}[\log(f_{bias}(\mathbf{h}))] + \mathbb{E}_{\mathbf{h} \sim p(\mathbf{h}|\hat{s}=0)}[\log(1 - f_{bias}(\mathbf{h}))]$$
$$= \sum_h p(\mathbf{h}|\hat{s}=1)\log(f_{bias}(\mathbf{h})) + \sum_h p(\mathbf{h}|\hat{s}=0)\log(1 - f_{bias}(\mathbf{h}))$$

If we fix $\theta_{class}$ and $\theta_{imp}$, then the optimal $f_{bias}$ is $\frac{p(\mathbf{h}|\hat{s}=1)}{p(\mathbf{h}|\hat{s}=1)+p(\mathbf{h}|\hat{s}=0)}$. By substituting the optimal $f_{bias}$ in Eq. 7 we get:

$$\min_{\theta_{class}} \max_{\theta_{imp}} \mathbb{E}_{\mathbf{h}\sim p(\mathbf{h}|\hat{s}=1)}[\log \frac{p(\mathbf{h}|\hat{s}=1)}{p(\mathbf{h}|\hat{s}=1)+p(\mathbf{h}|\hat{s}=0)}]+$$

$$\mathbb{E}_{\mathbf{h}\sim p(\mathbf{h}|\hat{s}=0)}[\log(1-\frac{p(\mathbf{h}|\hat{s}=1)}{p(\mathbf{h}|\hat{s}=1)+p(\mathbf{h}|\hat{s}=0)})]$$

We further simplify the objective function and get:

$$\mathbb{E}_{\mathbf{h}\sim p(\mathbf{h}|\hat{s}=1)}[\log \frac{p(\mathbf{h}|\hat{s}=1)}{p(\mathbf{h}|\hat{s}=1)+p(\mathbf{h}|\hat{s}=0)}]+$$

$$\mathbb{E}_{\mathbf{h}\sim p(\mathbf{h}|\hat{s}=0)}[\log \frac{p(\mathbf{h}|\hat{s}=0)}{p(\mathbf{h}|\hat{s}=1)+p(\mathbf{h}|\hat{s}=0)}]$$

$$=\sum_h p(\mathbf{h}|\hat{s}=1)\log \frac{p(\mathbf{h}|\hat{s}=1)}{p(\mathbf{h}|\hat{s}=1)+p(\mathbf{h}|\hat{s}=0)}+$$

$$\sum_h p(\mathbf{h}|\hat{s}=0)\log \frac{p(\mathbf{h}|\hat{s}=0)}{p(\mathbf{h}|\hat{s}=1)+p(\mathbf{h}|\hat{s}=0)}$$

$$=-\log 4+2JS(p(\mathbf{h}|\hat{s}=1);p(\mathbf{h}|\hat{s}=0))$$

By removing the constants, we can further simplify the min-max optimization with optimal $f_{bias}$ to:

$$\min_{\theta_{class}} \max_{\theta_{imp}} JS(p(\mathbf{h}|\hat{s}=1);p(\mathbf{h}|\hat{s}=0))$$

The optimal $f_{imp}$ thereby maximizes the following objective:

$$JS(p(\mathbf{h}|\hat{s}=1);p(\mathbf{h}|\hat{s}=0))$$

Since $\hat{y}=\sigma(\mathbf{h}.\mathbf{w})$, $f_{imp}$ generates imputation so that

$$\min_{\theta_{class}} \sup_{\hat{s}} JS(p(\hat{y}|\hat{s}=1);p(\hat{y}|\hat{s}=0))$$

The supremum value will make the two probabilities maximally different. Therefore, for the optimal $f_{imp}$ we get,

$$\min_{\theta_{class}} \sup_{\hat{s}} |p(\hat{y}|\hat{s}=1)-p(\hat{y}|\hat{s}=0)|$$

Let $\Delta DP_{\theta_{class},\theta_{imp}} = |p(\hat{y}=1|\hat{s}=1)-p(\hat{y}=1|\hat{s}=0)|$, then, the optimal $\theta^*_{imp}$ generates the worst-case (minimum fairness) imputation so that $\Delta DP_{\theta_{class},\theta^*_{imp}} \geq \Delta DP_{\theta_{class},\theta_{imp}}$ □

THEOREM 3

*Proof.* From Theorem 2, we get,

$$\min_{\theta_{class}} \sup_{\hat{s}} |p(\hat{y}|\hat{s}=1)-p(\hat{y}|\hat{s}=0)|$$

Let us assume that there can be $m$ different $\hat{s}$ that produce sensitive attributes $\hat{s}_1, \hat{s}_2, ..., \hat{s}_m$. If we let $D_i$ be the uniform distribution over each $s_i$, then BFtS minimizes the $|p(\hat{y}|\hat{s}=1)-p(\hat{y}|\hat{s}=0)|$ in the worst case over these different distributions $D_i$. Therefore, $f_{class}$ is a minimax estimator. The objective of Eq. 7 with optimal $\theta^*_{bias}, \theta^*_{imp}$ and $f^*_{class}$ becomes:

$$\inf_{\hat{y}} \sup_{\hat{s}} |p(\hat{y}|\hat{s}=1)-p(\hat{y}|\hat{s}=0)|$$

Therefore, $f_{class}$ is a minimax estimator and the maximal $\Delta DP$ of BFtS is minimum among all estimators of $s$.

□

At the global minima of $\theta_{class}$, $|p(\hat{y}|\hat{s}=1)-p(\hat{y}|\hat{s}=0)|$ will be minimum. Its minimum value is 0. Therefore, $p(\mathbf{h}|\hat{s}=1)-p(\mathbf{h}|\hat{s}=0)=0$. The optimal $\theta^*_{class}$ achieves demographic parity for the worst case.

COROLLARY 1

*Proof.* If $s'$ is a separate imputation independent of $f_{class}$ and $f_{bias}$, then the min max game according to Dai & Wang (2021) is the following:

$$\min_{\theta_{class}} \max_{\theta_{bias}} \mathbb{E}_{\mathbf{h} \sim p(\mathbf{h}|s'=1)}[\log f_{bias}(\mathbf{h})]$$
$$+ \mathbb{E}_{\mathbf{h} \sim p(\mathbf{h}|s'=0)}[\log(1 - f_{bias}(\mathbf{h}))].$$

With optimal $\theta_{bias}^*$, the optimization problem simplifies to-

$$\min_{\theta_{class}} -\log 4 + JS(p(\mathbf{h}|s'=1); p(\mathbf{h}|s'=0)).$$

With independent imputation $s'$, the adversary $f_{bias}$ tries to approximate the lower bound of Jensen Shannon Divergence (Weng, 2019). In BFtS, the adversarial imputation $f_{imp}$ tries to approximate the upper bound of the JS divergence, and the classifier $f_{class}$ tries to minimize the upper bound on the JS divergence provided by the adversarial imputation (see Proposition 1). Therefore,

$$JS((p(\mathbf{h}|s'=1); p(\mathbf{h}|s'=0))) \leq JS(p(\mathbf{h}|\hat{s}=1); p(\mathbf{h}|\hat{s}=0)).$$

$\square$

CONVERGENCE OF ADVERSARIAL LEARNING FOR THREE PLAYERS VS TWO PLAYERS WITH INDEPENDENT IMPUTATION

Let us consider a scenario when the independent missing sensitive value imputation (Dai & Wang, 2021) fails to converge. There are three possible reasons for the convergence to fail:

1. $s'$ is always 1
2. $s'$ is always 0
3. $s'$ is uniform, i.e. $p(\mathbf{h}|s'=1) = p(\mathbf{h}|s'=0)$

In scenario 1 and 2 the supports of $p(\mathbf{h}|s'=1)$ and $p(\mathbf{h}|s'=0)$ are disjoint. According to (Dai & Wang, 2021), the optimization for the classifier with an optimal adversary is

$$\min_{\theta_{class}} -\log 4 + 2JS(p(\mathbf{h}|s'=1); p(\mathbf{h}|s'=0))$$

As the supports of $p(\mathbf{h}|s'=1)$ and $p(\mathbf{h}|s'=0)$ are disjoint, $JS(p(\mathbf{h}|s'=1); p(\mathbf{h}|s'=0))$ is always 0. The gradient of the JS divergence vanishes, and the classifier gets no useful gradient information—it will minimize a constant function. This results in extremely slow training of the node classifier, and it may not converge.

In scenario 3, $\hat{s}$ is uniform. Therefore, $p(\mathbf{h}|s'=1)$ will be equal to $p(\mathbf{h}|s'=0)$ which results in $JS(p(\mathbf{h}|s'=1); p(\mathbf{h}|s'=0))$ being equal to 0. Moreover, the classifier may assign all training samples to a single class and yet have an adversary that may not be able to distinguish the sensitive attributes of the samples as $\hat{s}$ is random. If the classifier assigns all samples to the majority class, it will achieve low classification loss $\mathcal{L}_{class}$ along with minimum $\mathcal{L}_{bias}$. The training of the classifier may get stuck in this local minima (see Eq. 4). This phenomenon is similar to the mode collapse of GANs (Thanh-Tung & Tran, 2020).

Based on Corollary 1, our approach is more robust to the convergence issues described above.

TRAINING ALGORITHM

Algorithm 1 is a high-level description of the key steps applied for training BFtS. It receives the graph $\mathcal{G}$, labels $y$, labeled nodes $\mathcal{V}_L$, nodes with observed sensitive attributes $\mathcal{V}_S$ and hyperparameters, $\alpha$ and $\beta$ as inputs and outputs the GNN classifier $f_{class}$, sensitive attribute predictor, $f_{bias}$ and missing data imputation GNN $f_{imp}$. We first estimate $\hat{si}$ and then fix $\theta_{class}$ and $\theta_{bias}$ to update $\theta_{imp}$. Then we update $\theta_{bias}$ with Eq. 5. After that, we fix $\theta_{bias}$ and $\theta_{imp}$ and update $\theta_{class}$ with Eq. 4 and repeat these steps until convergence.

---

**Algorithm 1** Training BFtS

---

**Input:** $\mathcal{G}$; $y$; $\mathcal{V}_L$; $\mathcal{V}_S$; $\alpha$, $\beta$
**Output:** $f_{class}$, $f_{bias}$, $f_{imp}$

1: **repeat**
2:    Get the estimated sensitive attributes $\hat{si}$ with $f_{imp}$
3:    Fix $\theta_{class}$ and $\theta_{bias}$ and update $\theta_{imp}$ using Eq. 6
4:    Update $\theta_{bias}$ using Eq. 5
5:    Fix $\theta_{imp}$ and $\theta_{bias}$ and update $\theta_{class}$ using Eq. 4
6: **until** converge

---

## COMPLEXITY ANALYSIS

BFtS model consists of two GNNs and one DNN. If we consider GCN for training the GNN, the time complexity is $\mathcal{O}(L|V|F^2 + L|E|F)$, and space complexity is $\mathcal{O}(LE + LF^2 + L|V|F)$, where $L$ is the number of layers and $F$ is the nodes in each layer (We assume that the number of nodes in each layer is the same) (Blakely et al., 2021). The complexity is polynomial, and therefore, BFtS is scalable. The complexity of FairGNN (Dai & Wang, 2021) is the same as the complexity of BFtS. FairGNN makes an additional forward pass to the trained sensitive attribute imputation network. Therefore, the empirical running time of FairGNN is higher than the one for BFtS, as shown in the next section.

## RUNNING TIME

Table 2 shows that Debias has the lowest runtime but performs the worst in fairness (See Fig. 4 and 5). BFtS had the second-lowest runtime. While the 3-player network adds complexity, it removes the requirement of training a separate imputation method (as for other benchmarks). Figure 6 presents

| | German | Credit | Bail | NBA | pokec-z | pokec-n |
|---|---|---|---|---|---|---|
| Debias | **21.74** | **110.99** | **170.95** | **14.85** | **650.67** | **647.62** |
| FairGNN | 31.43 | 229.41 | 340.48 | 29.12 | 1356.57 | 1234.62 |
| FairVGNN | 212.64 | 10557.79 | 2554.72 | 234.57 | 15956.62 | 15632.75 |
| RNF | 136.67 | 432.31 | 655.54 | 138.31 | 1945.58 | 2012.13 |
| FairSIN | 615.74 | 15142.25 | 2884.34 | 342.14 | 20565.13 | 20456.34 |
| BFtS | 28.91 | 168.88 | 242.88 | 22.34 | 942.35 | 956.42 |

Table 2: Running time (secs) of different methods

the convergence behavior of $\mathcal{L}_{class}$, $\mathcal{L}_{bias}$ and $\mathcal{L}_{imp}$ across training epochs on the NBA dataset.

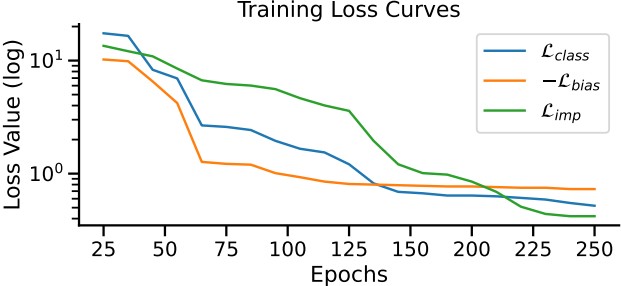

Figure 6: Convergence of $\mathcal{L}_{class}$, $\mathcal{L}_{bias}$ and $\mathcal{L}_{imp}$. All three losses converge smoothly, confirming theoretical stability.

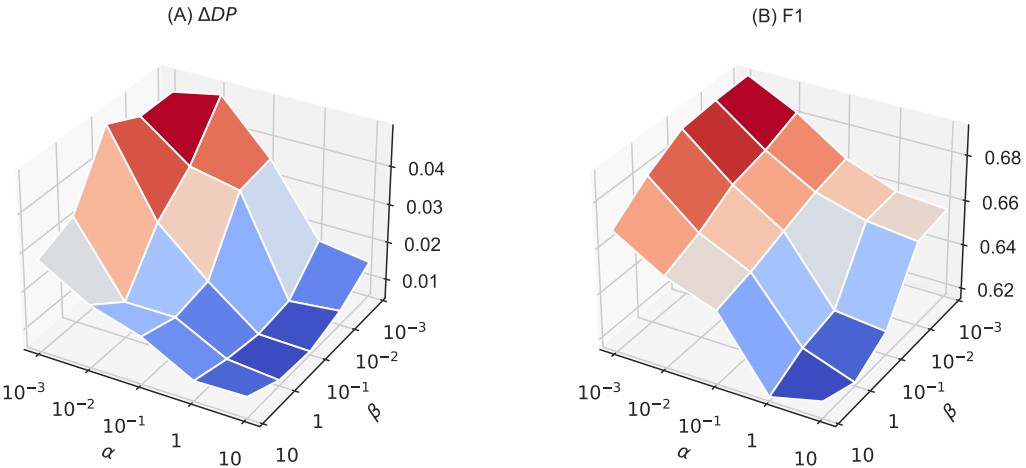

Figure 7: Sensitivity analysis of $\alpha$ and $\beta$. BFtS is more sensitive to $\alpha$ than $\beta$. These parameters can be optimized using automated techniques, such as grid search.

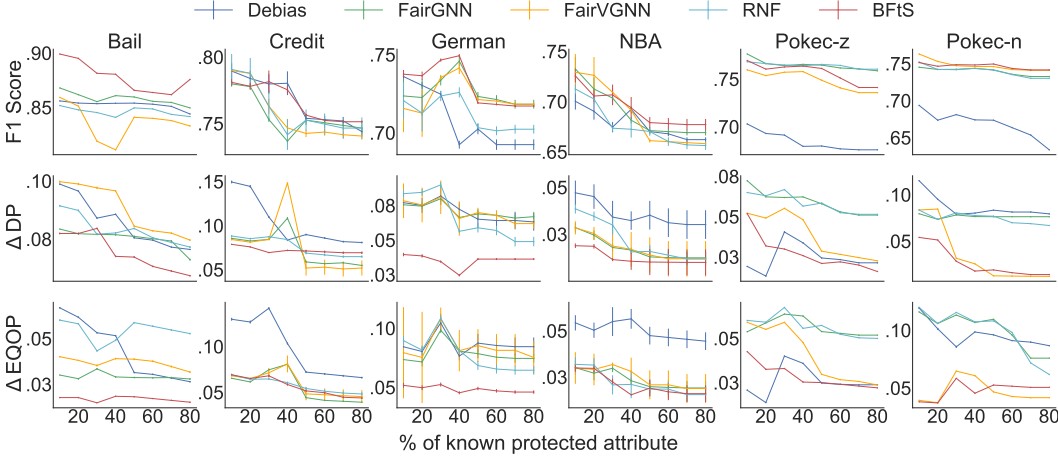

Figure 8: Performance of the models on different datasets for varying amounts of observed data $\mathcal{V}_S$. In the majority of the settings, our approach (BFtS) achieves a better fairness $\times$ accuracy trade-off than the baselines.

| | | VANILLA | DEBIAS | FairGNN | RNF | FairVGNN | FairSIN | BFtS (Ours) |
|---|---|---|---|---|---|---|---|---|
| BAIL | AVPR (↑) | 0.86±0.00 | 0.81±0.00 | 0.82±0.00 | 0.81±0.00 | 0.80 ±0.00 | 0.81±0.00 | **0.83±0.00** |
| | F1 score (↑) | 0.88±0.00 | **0.85 ±0.00** | 0.84±0.00 | 0.84±0.01 | 0.84 ±0.00 | 0.84±0.00 | **0.85±0.00** |
| | %ΔDP (↓) | 19.10±0.03 | 8.78±0.02 | 7.9±0.03 | 8.50±0.04 | 9.81 ±0.09 | 8.64 ±0.20 | **6.70±0.03** |
| | %ΔEQOP (↓) | 14.20±0.04 | 7.50±0.03 | 4.10±0.04 | 4.70±0.05 | 4.79±0.06 | 3.92 ±0.09 | **2.70±0.01** |
| | %ΔEQODDs (↓) | 11.49±0.08 | 8.61±0.02 | 5.28±0.08 | 3.42±0.07 | 5.01±0.02 | 4.96±0.01 | **2.81±0.02** |
| CREDIT | AVPR (↑) | 0.85±0.00 | 0.82±0.00 | **0.83±0.00** | 0.81±0.00 | 0.81 ±0.02 | 0.81 ±0.02 | 0.82±0.00 |
| | F1 score (↑) | 0.79±0.00 | 0.73 ±0.00 | **0.76±0.00** | 0.75±0.01 | 0.73 ±0.00 | 0.75 ±0.00 | **0.76±0.00** |
| | %ΔDP (↓) | 12.10±0.10 | 8.80±0.20 | 5.60±0.08 | 5.30±0.09 | 7.77 ±0.25 | **5.21 ±0.12** | 5.40±0.02 |
| | %ΔEQOP (↓) | 14.50±0.10 | 9.20±0.10 | 5.90±0.10 | 5.10±0.00 | 5.56 ±0.21 | 4.95 ±0.13 | **4.50±0.01** |
| | %ΔEQODDs (↓) | 12.30±0.20 | 10.10±0.20 | 6.03±0.09 | 5.20±0.03 | 5.98 ±0.24 | 5.30 ±0.11 | **4.85±0.02** |
| GERMAN | AVPR (↑) | 0.75±0.00 | 0.72 ±0.00 | 0.73±0.00 | 0.72±0.01 | 0.70 ±0.01 | 0.73 ±0.00 | **0.74±0.00** |
| | F1 score (↑) | 0.78±0.01 | 0.73 ±0.00 | 0.73±0.00 | 0.72±0.01 | 0.72±0.00 | 0.72 ±0.00 | **0.74±0.00** |
| | %ΔDP (↓) | 8.30±0.10 | 5.02±0.20 | 5.70 ±0.30 | 4.20±0.00 | 4.37 ±3.06 | 4.05 ±0.08 | **2.74±0.02** |
| | %ΔEQOP (↓) | 7.40±0.08 | 6.24±0.20 | 5.70±0.20 | 5.90±0.01 | 3.63 ±3.03 | 4.2 ±0.13 | **1.70±0.04** |
| | %ΔEQODDs (↓) | 7.78±0.01 | 6.91±0.30 | 5.91±0.08 | 6.26±0.08 | 3.75 ±1.13 | 5.01 ±0.08 | **2.14±0.03** |
| NBA | AVPR (↑) | 0.76±0.00 | 0.73±0.00 | 0.73±0.00 | 0.73±0.00 | 0.73 ±0.07 | 0.73 ±0.01 | **0.74±0.00** |
| | F1 score (↑) | 0.73±0.00 | 0.67 ±0.00 | 0.70±0.00 | 0.69±0.05 | 0.69 ±0.05 | 0.70 ±0.02 | **0.71±0.00** |
| | %ΔDP (↓) | 13.10±0.02 | 6.40±0.04 | 1.20±0.03 | 1.50±0.05 | 2.39 ±1.2 | 1.5 ±0.21 | **1.10±0.04** |
| | %ΔEQOP (↓) | 11.50±0.02 | 5.50±0.02 | 5.30±0.00 | 3.10±0.03 | 4.19 ±2.1 | 3.9 ±0.07 | **2.70±0.01** |
| | %ΔEQODDs (↓) | 12.10 ±0.3 | 8.51±0.2 | 4.23±0.10 | 4.92±0.03 | 5.62 ±0.91 | 5.12 ±0.02 | **2.90±0.05** |
| POKEC-Z | AVPR (↑) | 0.76±0.00 | 0.72±0.00 | 0.72±0.00 | **0.73±0.00** | 0.72 ±0.07 | 0.71 ±0.05 | **0.73±0.00** |
| | F1 score (↑) | 0.73±0.00 | 0.69 ±0.00 | 0.70±0.00 | 0.69±0.00 | 0.68 ±0.01 | **0.71 ±0.03** | **0.71±0.00** |
| | %ΔDP (↓) | 12.10±0.01 | 7.40±0.02 | 5.20±0.01 | 5.51±0.04 | 5.69 ±1.1 | 5.98 ±1.06 | **4.10±0.04** |
| | %ΔEQOP (↓) | 16.50±0.01 | 8.50±0.06 | 3.30±0.00 | 2.10±0.03 | 4.19 ±1.9 | 2.1 ±1.09 | **1.70±0.03** |
| | %ΔEQODDs (↓) | 13.10±0.04 | 7.59±0.02 | 3.18±0.08 | 2.81±0.04 | 5.09 ±1.02 | 2.56 ±0.57 | **1.91±0.04** |
| POKEC-N | AVPR (↑) | 0.74±0.00 | 0.71±0.00 | **0.72±0.00** | **0.72±0.00** | 0.70 ±0.08 | 0.71 ±0.02 | **0.72±0.00** |
| | F1 score (↑) | 0.75±0.00 | 0.70 ±0.00 | 0.71±0.00 | 0.71±0.00 | 0.70 ±0.03 | 0.72 ±0.04 | **0.73±0.00** |
| | %ΔDP (↓) | 11.10±0.02 | 5.40±0.01 | 2.10±0.01 | 2.50±0.01 | 2.39 ±1.7 | 2.12 ±0.98 | **1.89±0.02** |
| | %ΔEQOP (↓) | 10.50±0.01 | 5.50±0.02 | 3.10±0.00 | 2.90±0.01 | 3.19 ±2.2 | 2.1 ±0.85 | **1.80±0.01** |
| | %ΔEQODDs (↓) | 11.56±0.02 | 6.15±0.03 | 4.81±0.04 | 3.04±0.02 | 4.51 ±2.13 | 3.49 ±0.05 | **2.05±0.04** |

Table 3: AVPR, F1 score, $\%\Delta DP$, $\% \Delta$ EQOP, and $\% \Delta$ EQODDs of different methods. For the BAIL, and NBA dataset, we outperform all baselines in terms of accuracy and fairness. For other datasets, we improve fairness while slightly sacrificing accuracy.

| | | VANILLA | DEBIAS | FAIRGNN | RNF | FairVGNN | FairSIN | BFtS (Ours) |
|---|---|---|---|---|---|---|---|---|
| **BAIL** | *AVPR* (↑) | 0.88±0.00 | 0.81±0.00 | 0.81±0.00 | 0.82±0.00 | 0.82 ±0.01 | 0.83 ±0.01 | **0.84±0.00** |
| | *F1* (↑) | 0.71±0.00 | 0.59 ±0.00 | 0.68±0.00 | 0.62±0.003 | 0.66±0.01 | 0.69 ±0.02 | **0.70±0.00** |
| | %$\Delta DP$ (↓) | 11.10±0.04 | 12.70±0.04 | 8.20±0.03 | 9.80±0.05 | 8.48 ±0.51 | 8.31±0.03 | **7.9±0.04** |
| | %$\Delta EQOP$ (↓) | 8.90±0.01 | 6.50±0.06 | 4.30±0.03 | 5.90±0.01 | 5.88 ±0.08 | 3.96 ±0.21 | **2.80±0.02** |
| | %$\Delta EQODDs$ (↓) | 9.18±0.02 | 6.93±0.02 | 5.21±0.04 | 6.42±0.03 | 5.89 ±0.06 | 4.21 ±0.26 | **2.93±0.02** |
| **CREDIT** | *AVPR* (↑) | 0.79±0.00 | 0.73±0.00 | 0.75±0.00 | 0.75 ±0.01 | **0.76±0.00** | 0.74 ±0.00 | 0.74±0.00 |
| | *F1* (↑) | 0.71±0.00 | 0.68 ±0.00 | 0.69±0.00 | **0.71±0.00** | 0.70 ±0.00 | 0.70 ±0.00 | **0.71±0.00** |
| | %$\Delta DP$ (↓) | 14.20±0.02 | 9.10 ±0.02 | 8.02±0.05 | 6.40±0.04 | 6.70 ±0.05 | 5.01 ±0.20 | **4.20±0.02** |
| | %$\Delta EQOP$ (↓) | 18.30±0.20 | 15.12±0.08 | 12.15±0.10 | 10.95±0.06 | 9.75±0.09 | 9.31 ±0.09 | **8.60±0.04** |
| | %$\Delta EQODDs$ (↓) | 15.51±0.10 | 10.22±0.01 | 9.04±0.32 | 9.92±0.05 | 8.96±0.02 | 8.54 ±0.10 | **7.57±0.02** |
| **GERMAN** | *AVPR* (↑) | 0.73±0.00 | 0.70 ±0.00 | 0.71±0.00 | **0.72±0.00** | 0.71±0.00 | 0.70 ±0.00 | 0.71±0.00 |
| | *F1* (↑) | 0.74±0.00 | 0.71 ±0.00 | 0.70±0.00 | 0.72±0.01 | 0.71 ±0.00 | 0.71 ±0.01 | **0.72±0.00** |
| | %$\Delta DP$ (↓) | 9.80±0.07 | 6.12±0.06 | 6.70 ±0.10 | 7.10±0.00 | 6.01±0.06 | 5.01 ±0.02 | **4.1±0.04** |
| | %$\Delta EQOP$ (↓) | 10.80±0.04 | 6.70±0.10 | 5.80±0.04 | 6.10±0.06 | 5.20±0.07 | 4.13 ±0.09 | **3.90±0.04** |
| | %$\Delta EQODDs$ (↓) | 11.15±0.05 | 6.87±0.20 | 6.13±0.05 | 5.89±1.06 | 4.97±0.03 | 4.76 ±0.02 | **3.21±0.05** |
| **NBA** | *AVPR* (↑) | 0.74±0.00 | 0.70±0.00 | 0.71±0.00 | **0.72±0.00** | 0.71±0.00 | 0.70 ±0.00 | 0.71±0.00 |
| | *F1* (↑) | 0.79±0.00 | 0.75 ±0.00 | 0.75±0.00 | 0.75±0.00 | 0.75±0.01 | 0.74 ±0.08 | **0.76±0.00** |
| | %$\Delta DP$ (↓) | 7.60±0.03 | 4.20±0.03 | 2.30±0.01 | 3.10±0.04 | 2.10±0.04 | 3.19 ±0.10 | **2.01±0.01** |
| | %$\Delta EQOP$ (↓) | 11.20±0.03 | 5.40±0.04 | 3.30±0.01 | 3.30±0.01 | 3.12±0.05 | 2.91 ±0.05 | **2.10±0.01** |
| | %$\Delta EQODDs$ (↓) | 10.21±0.04 | 6.51±0.02 | 3.65±0.07 | 3.95±0.02 | 3.48±0.08 | 2.85 ±0.02 | **1.89±0.02** |
| **POKEC-Z** | *AVPR* (↑) | 0.78±0.00 | 0.73±0.00 | **0.74±0.00** | 0.73±0.00 | 0.73 ±0.07 | 0.74 ±0.05 | **0.74±0.00** |
| | *F1 score* (↑) | 0.75±0.00 | 0.71 ±0.00 | 0.72±0.00 | 0.72±0.00 | 0.73 ±0.01 | 0.73 ±0.03 | **0.74±0.00** |
| | %$\Delta DP$ (↓) | 10.19±0.07 | 8.21±0.08 | 5.98±0.07 | 5.18±0.08 | 5.79 ±1.08 | 4.98 ±2.01 | **3.90±0.05** |
| | %$\Delta EQOP$ (↓) | 12.13±0.01 | 6.85±0.02 | 2.98±0.02 | 2.38±0.07 | 3.15 ±2.01 | 2.90 ±1.07 | **1.98±0.01** |
| | %$\Delta EQODDs$ (↓) | 13.13±0.02 | 6.93±0.06 | 3.15±0.07 | 2.95±0.03 | 2.65 ±1.01 | 2.13 ±0.97 | **1.79±0.02** |
| **POKEC-N** | *AVPR* (↑) | 0.75±0.00 | 0.72±0.00 | 0.72±0.00 | 0.72±0.00 | 0.71±0.08 | 0.72 ±0.04 | **0.73±0.00** |
| | *F1 score* (↑) | 0.78±0.00 | 0.75 ±0.00 | 0.75±0.00 | 0.74±0.00 | 0.74 ±0.02 | 0.75 ±0.01 | **0.76±0.00** |
| | %$\Delta DP$ (↓) | 14.10±0.01 | 6.51±0.02 | 3.40±0.02 | 4.12±0.02 | 2.98 ±1.9 | 2.75 ±1.2 | **1.89±0.02** |
| | %$\Delta EQOP$ (↓) | 12.50±0.07 | 5.50±0.02 | 3.10±0.00 | 2.90±0.01 | 3.19 ±2.2 | 2.1 ±0.85 | **1.80±0.01** |
| | %$\Delta EQODDs$ (↓) | 13.57±0.03 | 5.91±0.04 | 3.70±0.01 | 3.20±0.01 | 2.91±1.03 | 2.43 ±0.06 | **1.71±0.05** |

Table 4: AVPR, F1, % $\Delta$DP, % $\Delta$ EQOP, and % $\Delta$ EQODDs of different methods. GAT is the GNN architecture for all models. For the BAIL dataset, our model outperforms every other baseline in terms of fairness and accuracy. For the GERMAN, NBA and CREDIT datasets, we perform more fairly but less accurately.

IMPUTATION ACCURACY

Table 5 shows the imputation accuracy of the methods. We exclude results for Debias, which is based only on the sensitive information available, and for FairVGNN and FairSIN, as they are identical to FairGNN (same imputation method). The results show that BFtS imputation outperforms independent imputation methods. BFtS worst-case imputation based on the LDAM loss outperforms the alternatives in terms of accuracy. This is due to the adversarial missingness process where low-degree nodes are selected to have missing values. Independent imputation methods are less effective in this adversarial setting.

|         | German | Credit | Bail | NBA  | pokec-z | pokec-n |
|---------|--------|--------|------|------|---------|---------|
| FairGNN | 0.70   | 0.90   | 0.56 | **0.78** | 0.81    | 0.84    |
| RNF     | 0.59   | 0.73   | 0.53 | 0.64 | 0.65    | 0.68    |
| BFtS    | **0.72** | **0.94** | **0.63** | **0.78** | **0.83** | **0.87** |

Table 5: Accuracy of missing sensitive imputation.

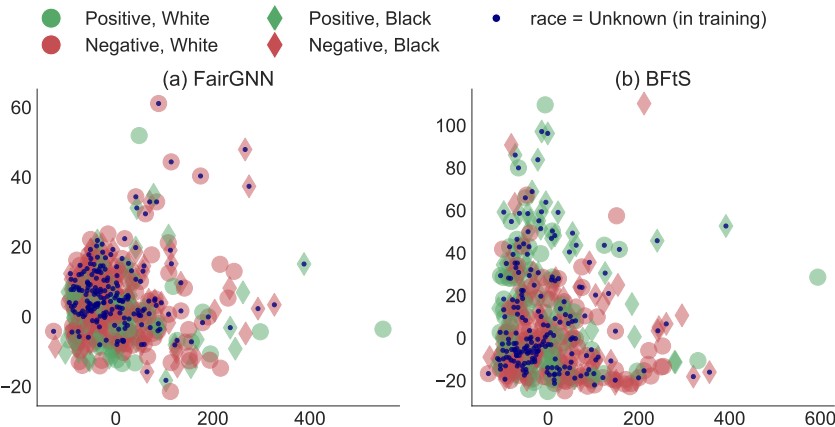

Figure 9: 2-D kernel PCA node representations for the BAIL dataset generated by the FairGNN and BFtS (our approach). We use different markers for nodes depending on their predicted class, sensitive attribute (race), and whether the sensitive attribute is missing. The results show that representations for nodes with missing values are more biased for FairGNN where they are concentrated in a negative class region. On the other hand, nodes with missing values are better spread over the space for BFtS representations. In BFtS, there are more 'Race = Black' nodes than in FairGNN that have missing sensitive values in training but are predicted to be positive.

WORST-CASE FAIRNESS ACCURACY TRADE-OFF

As our proposed model operates under a worst-case fairness assumption, it may overestimate the bias in the complete data, as illustrated in Figure 2 using the BAIL dataset. This results in a trade-off between fairness and utility, which is governed by a hyperparameter $\beta$. Figures 10(a), (b), and (c) show the F1 score, $1 - \Delta\text{DP}$, and $1 - \Delta\text{EQOP}$, respectively, as $\beta$ varies and with $30\%$ of sensitive values observed. Here, F1 serves as the utility metric, while $1 - \Delta\text{DP}$ and $1 - \Delta\text{EQOP}$ quantify fairness. We compare our BFtS model against a fair adversarial model trained with complete sensitive information. By tuning $\beta$, we balance predictive utility and fairness under worst-case bias, achieving an effective trade-off.

ADDITIONAL EXPERIMENTS

LARGE SCALE GRAPH DATASET

To verify the performance of BFtS on large-scale dataset, we generated a synthetic graph with 250k nodes and 100 features, as we were unable to identify a large-scale real-world graph dataset with

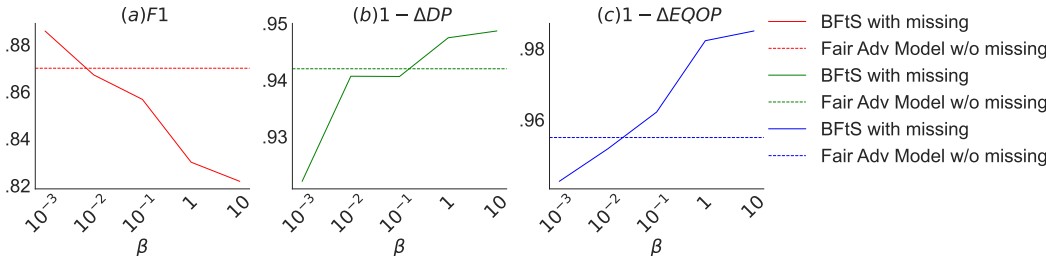

Figure 10: Fairness–utility trade-off of BFtS on the BAIL dataset as $\beta$ varies, compared to a fair adversarial model trained with complete data. Lower $\beta$ yields higher accuracy but lower fairness.

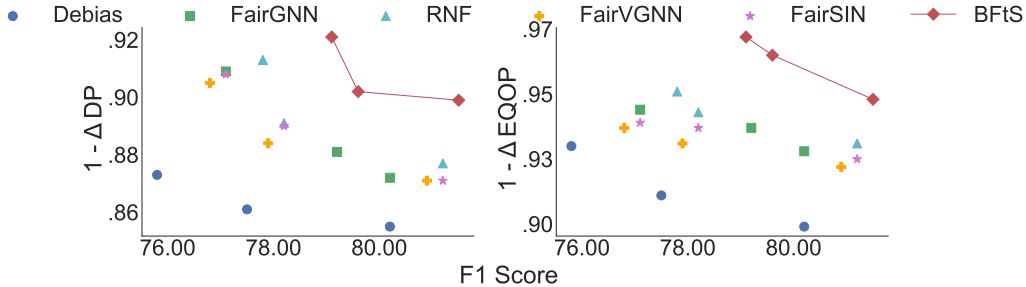

Figure 11: Fairness accuracy trade off for large scale graph dataset. BFtS achieves better fairness-accuracy trade-off

sensitive values. Following the setup in Figure 4 (with 150k nodes in the non-protected group and assortativity = 0.77). We compared BFtS against other baselines in Figure 11 across different hyperparameter settings. BFtS completed training in 85.4 minutes versus RNF's (the strongest baseline in this scenario) 114.6 minutes.

AVERAGE PERFORMANCE WITH GCN AND GAT

Table 3 shows the average results for BAIL, CREDIT, GERMAN, POKEK-Z, POKEK-N, and NBA datasets. In terms of accuracy and fairness, BFtS outperforms all baseline approaches for the POKEK-Z, POKEK-N, BAIL, GERMAN and NBA datasets. We also outperform competitors on the CREDIT dataset in terms of fairness, with a slight AVPR loss. Although we lose some AVPR in the CREDIT dataset, we win in F1 and the AVPR loss is negligible when compared to the fairness benefit. Table 4 shows the average results for BAIL, CREDIT, GERMAN, POKEK-Z, POKEK-N, and NBA datasets using GAT Velickovic et al. (2018) to train $f_{class}$ and $f_{imp}$. In terms of accuracy and fairness, BFtS outperforms all baseline approaches for the POKEK-Z, POKEK-N,BAIL dataset. We also outperform competitors on the CREDIT, GERMAN, and NBA datasets in terms of fairness with a slight AVPR loss. The AVPR loss is negligible when compared to the fairness benefit.

VISUALIZATION OF REPRESENTATIONS

Figure 9 shows kernel PCA node representations produced by FairGNN and BFtS using the BAIL dataset. We show the samples with missing "Race" with a navy dot in the middle and samples predicted as positive class and negative class in green and red, respectively. The representations generated by FairGNN are noticeably more biased than the ones generated by our approach. A higher number of "Race = Black" samples with missing sensitive values in training are predicted as positive by BFtS than for FairGNN. Moreover, BFtS spreads nodes with missing values more uniformly over the space. This illustrates how the missing value imputation of FairGNN underestimates the bias in the training data and, therefore, the model could not overcome the bias in the predictions.

| Method | MSE ($\pm$) | $\Delta$DP ($\pm$) |
|--------|-------------|--------------------|
| BFtS | $0.71 \pm 0.10$ | $0.15 \pm 0.02$ |
| Vanilla | $0.65 \pm 0.10$ | $0.21 \pm 0.01$ |

Table 6: Fair regression: MSE and Fairness ($\Delta DP$) Comparison Between BFtS and Vanilla Models

| | CROSS ENTROPY $f_{imp}$ | | | | LDAM $f_{imp}$ | | | |
|---|---|---|---|---|---|---|---|---|
| | AVPR ($\uparrow$) | F1 ($\uparrow$) | %$\Delta$DP ($\downarrow$) | %$\Delta$EQOP ($\downarrow$) | AVPR($\uparrow$) | F1($\uparrow$) | %$\Delta$DP ($\downarrow$) | %$\Delta$EQOP ($\downarrow$) |
| GERMAN | **0.75 ±0.00** | **0.73 ±0.00** | 4.10 ±0.05 | 3.70 ±0.03 | 0.74 ±0.00 | 0.74 ±0.00 | **2.74 ±0.02** | **1.7 ±0.04** |
| CREDIT | **0.83 ±0.00** | **0.77 ±0.01** | 5.55 ±0.03 | 4.70 ±0.05 | **0.83 ±0.00** | 0.76 ±0.00 | **5.4 ±0.02** | **4.5 ±0.01** |
| BAIL | **0.85 ±0.00** | **0.85 ±0.00** | 7.10 ±0.02 | 3.1 ±0.06 | 0.83 ±0.00 | **0.85 ±0.00** | **6.70 ±0.03** | **2.7 ±0.01** |
| NBA | **0.75 ±0.00** | **0.72 ±0.00** | 2.11 ±0.03 | 3.5 ±0.02 | 0.74 ±0.00 | 0.71 ±0.00 | **1.10 ±0.04** | **2.70 ±0.01** |
| SIMULATION | **0.94 ±0.00** | **0.96 ±0.00** | 16.00 ±0.03 | 8.00 ±0.08 | 0.92 ±0.00 | **0.96 ±0.01** | **12.00 ±0.03** | **5.10 ±0.04** |

Table 7: Ablation study using LDAM loss and cross-entropy loss for $f_{imp}$. Using LDAM loss gives a better fairness and accuracy trade-off.

### FAIR NODE REGRESSION

While fairness-aware node regression datasets are lacking, we created a synthetic node regression task (assortativity = 0.77) following our setup in Figure 5, with group-wise targets from $\mathcal{N}(0, 1)$ and $\mathcal{N}(2, 1)$. The Table 6 reports MSE and $\Delta DP$ (Berk et al., 2017) for regression.

### ABLATION STUDY

To see the impact of $\alpha$ and $\beta$, we train BFtS on the GERMAN dataset with different values of $\alpha$ and $\beta$. We consider values between $[10^{-3}, 10^{-2}, 10^{-1}, 1, 10]$ for both $\alpha$ and $\beta$. Figure 7 shows the $\Delta$DP and F1 of BFtS on the GERMAN dataset. $\alpha$ and $\beta$ control the impact of the adversarial loss $\mathcal{L}_A$ on the GNN classifier and the missing value imputation GNN, respectively. The figure shows that $\alpha$ has a larger impact than $\beta$ on the fairness and accuracy of the model.

We also vary the amount of observed data $\mathcal{V}_S$ and plot the results in Figure 8. We use $10\%, 20\%, ..., 80\%$, of training nodes as $\mathcal{V}_S$. For nearly all models, fairness increases and accuracy decreases with $|\mathcal{V}_S|$. With BFtS, both accuracy and bias often decline with the increase of $\mathcal{V}_S$. In the majority of the settings, our approach (BFtS) achieves a better fairness $\times$ accuracy trade-off than the baselines.

To see the impact of LDAM loss, we train $f_{imp}$ with cross-entropy loss and compare the performance with $f_{imp}$ trained with LDAM loss in Table 7. Evidently, LDAM achieves a better trade-off between accuracy and fairness.

