# OpenReview forum: "Fair Graph Machine Learning under Adversarial Missingness Processes"
_ICLR.cc/2026/Conference — ICLR 2026 Poster_

### Official Review · Reviewer_53p8 · 2025-10-29

**Soundness:** 3
**Presentation:** 2
**Contribution:** 3
**Rating:** 6
**Confidence:** 4

**Summary:**

This paper addresses a critical and underexplored problem in fair graph machine learning: the effect of adversarial missingness of sensitive attributes on fairness-aware Graph Neural Networks (GNNs). The authors argue that prior fairness methods assume that sensitive attributes are either fully available or missing completely at random (MCAR), which is unrealistic in practice. To overcome this limitation, the paper introduces Better Fair than Sorry (BFtS), a 3-player adversarial framework that jointly learns a node classifier, a fairness adversary that predicts sensitive attributes from learned embeddings, and an imputation adversary that imputes missing sensitive attributes to approximate the worst-case fairness scenario. Moreover, the authors provide theoretical analysis showing that BFtS corresponds to a min–max optimization that minimizes classifier bias under worst-case imputations.

**Strengths:**

1. This paper has principled theoretical grounding, clear definitions, and basic robust justification.
2. This paper proposes an innovative framework with 3-player adversarial setup elegantly unifies imputation, fairness estimation, and classification.
3. Extensive experiments on both synthetic and real-world benchmarks and the robustness is shown under varying degrees of missingness.

**Weaknesses:**

1. The 3-player adversarial training could be computationally heavy, particularly on large-scale graphs. A detailed runtime or memory comparison would strengthen the empirical analysis.
2. The study focuses on Demographic Parity (ΔDP) and Equality of Opportunity (ΔEQOP). It would be beneficial to include other fairness notions (e.g., Equalized Odds, Counterfactual Fairness) for completeness.
3. While the adversarial imputation approach is conceptually powerful, there is little discussion of how it performs under realistic partial observability (e.g., when only 5–10% of sensitive data is known).
4. The intuition behind the imputation adversary’s learned behavior could be further explored, perhaps via visualization or sensitivity analysis.

**Questions:**

1. How sensitive is BFtS to hyperparameter tuning, especially the fairness weight (α) and imputation adversary weight (β)? Could the authors provide empirical or theoretical guidance on selecting them?
2. Does the degree-bias assumption for adversarial missingness generalize to graphs with highly non-homophilous structures?
3. Could BFtS be extended to handle multi-valued or continuous sensitive attributes rather than binary ones?
4. Is there any evidence of mode collapse or instability during the 3-player adversarial training, and if so, how is it mitigated?

---

> ### Author Response · Authors · 2025-11-21
> **Rebuttal by Authors**
>
> We thank the reviewer for their thoughtful evaluation of our work and appreciate their positive comments on the motivation and contributions of BFtS. The reviewer’s main concerns involve the computational overhead of the 3-player framework, performance under very limited observability, and the behavior of the imputation adversary. We address each of these points below. If you find our response satisfactory, we would be grateful if you could update your score. Please let us know if you have any further questions.
>
> > The 3-player adversarial training could be … would strengthen the empirical analysis..
>
> BFtS’s complexity depends on the underlying GNN. For GCN, we provide a time complexity analysis in the Appendix: $\mathcal{O}(L|V|F^2 + L|E|F)$, where $L$, $|V|$, $F$, and $|E|$ denote the number of layers, nodes, features, and edges. We also add the space complexity (line 1109): $\mathcal{O}(L|E| + LF^2 + L|V|F)$.
> While BFtS is theoretically scalable, real-world large-scale graphs with sensitive attributes are unavailable, so we constructed a synthetic graph with 250k nodes and 100 features. Using the same setup as Figure 5, we compared BFtS with baselines (Appendix Figure 11). BFtS trained in 85.4 minutes, outperforming the second-best baseline, RNF, which took 114.6 minutes. We now reference this experiment in Section 5 (line 474).
>
> > The study focuses on Demographic Parity … (e.g., Equalized Odds, Counterfactual Fairness) for completeness.
>
> In Tables 3 and 4 in Appendix A, we add additional results of Equalized Odds [1]  for all datasets and baselines. Due to time constraints, we could not add counterfactual fairness. We will add them in the final version.
>
> > While the adversarial imputation approach is conceptually powerful, … partial observability (e.g., when only 5–10% of sensitive data is known).
>
> In Figure 8 of the Appendix, we have increased the range and currently vary % of missing values from 20% to 90%. BFtS outperforms other baselines in the majority of the settings in terms of fairness-accuracy trade-off. The experiments show that our approach excels in high missingness scenarios such as the one mentioned by the reviewer.
>
> > The intuition behind the imputation adversary’s learned behavior ... visualization or sensitivity analysis.
>
> We have the following experiments in the appendix and have now referred to these experiments in the main paper at line 475:
> - Figure 9 (Appendix) visualizes the representations learned by FairGNN and BFtS on the Bail dataset. FairGNN’s representations exhibit noticeably higher bias, while BFtS correctly predicts more “Race = Black” samples with missing sensitive values as positive.
> - We also analyze the sensitivity of $\alpha$ and $\beta$ in the Appendix (Fig. 7), which shows that BFtS is more sensitive to $\alpha$ than $\beta$.
> - We further investigate the effect of the worst-case assumption by varying only $\beta$ and report results on the Bail dataset in Fig. 10 (Appendix). Comparing BFtS to a fair adversarial model trained with complete sensitive attributes, we show that tuning $\beta$ provides a controllable trade-off between accuracy and worst-case fairness.
>
>  > How sensitive is BFtS to hyperparameter tuning, … selecting them?
>
> We show a hyperparameter sensitivity study with $\alpha$  and $\beta$ in Appendix (Fig. 7). BFtS is more sensitive to $\alpha$ than $\beta$. We have now referred to this result in Section 5 line 475.
>
> > Does the degree-bias assumption ... non-homophilous structures?
>
> [2] and [3] empirically show that degree bias exists in non-homophilic graphs, such as Chameleon and Squirrel [4]. We have modified line 251 of our paper as ‘The degree bias assumption has been supported by both theoretical and empirical results in the literature for both homophilic and non-homophilic graphs’. Thank you for pointing this out.
>
> >Could BFtS be extended to handle multi-valued or continuous sensitive attributes rather than binary ones?
>
> For simplicity, we consider a binary sensitive attribute. However, BFtS can be extended to non-binary or continuous sensitive attributes by appropriately generalizing the imputation loss $\mathcal{L}\_{imp}$ and the bias loss $\mathcal{L}\_{bias}$ to multi-class or continuous settings. We have added this clarification in the paper at line 383.

---

> > ### Author Response · Authors · 2025-11-21
> > **Continued rebuttal**
> >
> > >Is there any evidence evidence of mode collapse ... how is it mitigated?
> >
> > Thank you for pointing this out. We theoretically prove that BFtS is more robust to mode collapse than two-player adversarial networks (Corollary 1). To further examine the convergence, we have plotted the training loss curves for $\mathcal{L}\_{class}$, $\mathcal{L}\_{bias}$, and $\mathcal{L}\_{imp}$ in Figure 6 of the Appendix.
> >
> > [1] Grant, David Gray. "Equalized odds is a requirement of algorithmic fairness." Synthese 2023.
> >
> > [2] Liu, Zemin, Trung-Kien Nguyen, and Yuan Fang. "On generalized degree fairness in graph neural networks." AAAI 2023.
> >
> > [3] Ju, Mingxuan, et al. "Graphpatcher: mitigating degree bias for graph neural networks via test-time augmentation." NeurIPS (2023).
> >
> > [4] Zhu et al. Beyond Homophily in Graph Neural Networks: Current Limitations and Effective Designs. NeurIPS (2020)

---

> > > ### Comment · Reviewer_53p8 · 2025-11-26
> > > **Official Comment by Reviewer 53p8**
> > >
> > > Thank you for the authors' response. I will keep my score unchanged.

---

> > > > ### Author Response · Authors · 2025-11-26
> > > > **Response to reviewer**
> > > >
> > > > Thanks for your response to our rebuttal. Please let us know if there is any additional question or weakness that we have not addressed in our rebuttal and that justifies keeping the original score. We believe we have addressed all the weaknesses and questions listed but we still have more time left for further discussion.

---

### Official Review · Reviewer_8hmM · 2025-10-31

**Soundness:** 3
**Presentation:** 2
**Contribution:** 3
**Rating:** 4
**Confidence:** 2

**Summary:**

This paper studies graph fairness when demographic attributes are partially observed under adversarial missingness processes.
It introduces BFtS (Better Fair than Sorry), a 3-player adversarial learning framework combining a GNN classifier, a fairness adversary, and an imputation adversary. BFtS imputes worst-case sensitive attributes to make fairness evaluation robust under adversarial missingness. Both theoretical analyses and empirical results demonstrate superior tradeoffs between fairness and accuracy compared to baselines.

**Strengths:**

- The work identifies a realistic yet overlooked issue: adversarial missingness of sensitive attributes, which can mislead fairness evaluations in graph learning, and formalizes two adversarial missingness problems (AMAFC, AMADB)

- Theorems 2 and 3 clearly demonstrate that BFtS minimizes worst-case bias and approximates robust fairness

- Extensive evaluations are conducted across synthetic and real-world datasets. Empirical results consistently show superior fairness–accuracy trade-offs and robustness under limited or missing sensitive data

- The paper is well organized and easy to follow

**Weaknesses:**

- The practicality of adversarial missingness is vague. Whether a value is missing or not seems to be difficult to control by adversaries. If the adversaries can deliberately drop some values, in this case, modifying these values seems to be a stronger adversary we can think about. It would be helpful to add more discussions on the practical scenarios of adversarial missingness in real-world cases.

- It would be helpful to add an introduction of the threat model setting in the main text.

- The bilevel optimization raises concerns about training stability. Although the authors propose the loss curve in Figure 6, it seems that Figure 6 is plotted with only a few points. It would be helpful if the training stability could be better shown in the experiments.

- It seems that the proportion of missing sensitive attributes is more than 50% of the total nodes in Figure 8. It would be helpful to see more experimental results under more stealthy attack settings.

**Questions:**

Please see Weaknesses

---

> ### Author Response · Authors · 2025-11-21
> **Rebuttal by Authors**
>
> We thank the reviewer for their thoughtful evaluation of our work, including the recognition of our problem formulation, theoretical guarantees, and empirical robustness. The reviewer’s main concerns relate to the practicality of adversarial missingness, the threat model description, training stability, and evaluations under more stealthy missingness attacks. We address each of these points below and in our paper. If our responses satisfactorily resolve these concerns, we would be grateful if you could update your score. Please let us know if you have any further questions.
>
> > The practicality of adversarial missingness is vague. … practical scenarios of adversarial missingness in real-world cases.
>
> As described in line 177: ``we focus on how the missingness process (i.e., the process that generates missing values) can lead to (intended or unintended) biases in a fair model trained using the imputed data, leading fair models to underestimate bias and remain unfair relative to the complete data”. In the case where an ill-intentioned agent wants to increase bias, we claim that interventions on the missingness process are easier to implement and harder to trace than data modifications during an auditing process. Directly altering sensitive values typically produces detectable inconsistencies across logs, backups, and correlated attributes and requires privileged write access that is tightly monitored. In contrast, inducing missingness through dropped fields, corrupted storage, or suppressed logging resembles ordinary pipeline failures and does not create possibly conflicting records. [1,2] also mention that when the data can be audited for correctness (e.g., it is cryptographically signed by its source), the adversarial perturbation mechanism is invalidated, and therefore, they also focus on adversarial missingness in causal structure learning (not missing value imputation or fairness). We have added a paragraph in the related work (line 139) highlighting this.
>
>
> > It would be helpful to add an introduction of the threat model setting in the main text.
>
> Thank you for the suggestion. We have added the following paragraph in Section 3 at line 181 of our paper.
>
> In our threat model, the asset is the graph dataset $\mathcal{G(\mathcal{V}, \mathcal{E}, \mathcal{X}, \mathcal{S})}$, and the threat is to increase the bias of a node classification model $f\_{class}$ applied to $\mathcal{G}$. The vulnerability arises because the adversary can choose a subset of sensitive attributes $\mathcal{V} \setminus \mathcal{V}\_S$ to hide, thereby inducing a strategically biased missingness pattern. Our mitigation is the BFtS framework, which jointly performs missing-data imputation and fair node classification to counter such adversarial missingness. The attacker’s capabilities and constraints are described next in the adversarial missingness formulations introduced in this section.
>
>
> > The bilevel optimization raises concerns about training stability. … It would be helpful if the training stability could be better shown in the experiments.
>
> Thank you for pointing this out. We have used adversarial learning (without bilevel optimization) for training BFtS. We prove robustness of BFtS to mode collapse and vanishing gradients (Corollary 1), two phenomena known to hinder the convergence of adversarial training and prevent min–max procedures from reaching Nash equilibria [3, 4]. We have added more points to Figure 6 for better visualization of loss convergence. These results directly support the practical stability of BFtS.
>
>
> > It seems that the proportion of missing sensitive attributes is more than 50% …  under more stealthy attack settings.
>
> In Figure 8 of the Appendix, we have increased the range and currently vary % of missing values from 20% to 90%. With more stealthy attacks (fewer than 50% missing values), the performance of other baselines also improves. Yet, BFtS achieves a better fairness accuracy trade-off than other baselines.
>
> [1] Koyuncu, Deniz, et al. "Deception by omission: Using adversarial missingness to poison causal structure learning." Proceedings of the 29th ACM SIGKDD Conference on Knowledge Discovery and Data Mining. 2023.
>
> [2] Koyuncu, Deniz, et al. "Adversarial missingness attacks on causal structure learning." ACM Transactions on Intelligent Systems and Technology 15.6 (2024): 1-60.
>
> [3] Zhang, Zhaoyu, Changwei Luo, and Jun Yu. "Towards the gradient vanishing, divergence mismatching and mode collapse of generative adversarial nets." Proceedings of the 28th ACM International Conference on Information and Knowledge Management. 2019.
>
> [4] Kreps, David M. "Nash equilibrium." The new Palgrave dictionary of economics. Palgrave Macmillan, London, 2018. 9251-9258.

---

### Official Review · Reviewer_1oQi · 2025-11-01

**Soundness:** 3
**Presentation:** 3
**Contribution:** 3
**Rating:** 6
**Confidence:** 3

**Summary:**

This paper investigates the problem of fair graph learning when sensitive attributes are missing under an adversarial missingness process. The authors propose Better Fair than Sorry (BFtS), a three-player adversarial framework involving a graph classifier, a bias discriminator, and a missing-value imputer. The method aims to enhance fairness robustness by simulating worst-case imputations. Both theoretical arguments and empirical evaluations are presented, demonstrating that BFtS achieves superior fairness–accuracy trade-offs on multiple graph datasets compared to existing methods.

**Strengths:**

1. Novel and important problem
The paper targets a realistic setting where sensitive attributes are not missing at random, which is often overlooked in existing fair graph learning literature. Formulating this as an adversarial missingness problem is both intuitive and practically meaningful.

2. Methodological soundness
The three-player adversarial design is conceptually well-motivated and integrates ideas from fairness, robust optimization, and adversarial learning into a unified framework. The training procedure is clearly described and the objectives are well defined.

3. Comprehensive experiments
The evaluation covers both synthetic and real-world datasets, reporting multiple fairness and accuracy metrics. The results consistently show that BFtS outperforms baseline methods under different missingness settings.

**Weaknesses:**

1. Limited theoretical depth
The theoretical results provide general insights but remain high-level. The proofs are brief and do not include convergence or generalization guarantees for the proposed min–max training objective. A more formal analysis of the optimization dynamics would strengthen the paper.

2. Comparison to related methods
While the paper compares BFtS with existing fair graph learning approaches, it could more clearly articulate how its mechanism differs from other fairness-aware imputation or robustness frameworks. The conceptual novelty may appear incremental without deeper discussion.

3. Experimental diversity
The adversarial missingness is modeled primarily through degree bias, which may not capture all possible real-world scenarios. Including other structural or attribute-based missingness patterns would make the empirical evaluation more convincing.

4. Fairness metrics and discussion
The choice of fairness metrics (Demographic Parity and Equality of Opportunity) is standard, but the paper could briefly justify why these particular measures were selected and whether the method generalizes to others.

5. Presentation details
Some notations are inconsistent between equations, and the visual presentation of a few figures could be improved for clarity.

**Questions:**

1. Limited theoretical depth
The theoretical results provide general insights but remain high-level. The proofs are brief and do not include convergence or generalization guarantees for the proposed min–max training objective. A more formal analysis of the optimization dynamics would strengthen the paper.

2. Comparison to related methods
While the paper compares BFtS with existing fair graph learning approaches, it could more clearly articulate how its mechanism differs from other fairness-aware imputation or robustness frameworks. The conceptual novelty may appear incremental without deeper discussion.

3. Experimental diversity
The adversarial missingness is modeled primarily through degree bias, which may not capture all possible real-world scenarios. Including other structural or attribute-based missingness patterns would make the empirical evaluation more convincing.

4. Fairness metrics and discussion
The choice of fairness metrics (Demographic Parity and Equality of Opportunity) is standard, but the paper could briefly justify why these particular measures were selected and whether the method generalizes to others.

---

> ### Author Response · Authors · 2025-11-21
> **Rebuttal by Authors**
>
> We thank the reviewer for their comments and positive assessment of our work and for recognizing its novelty, methodological soundness, and strong empirical evaluation. The main concerns of the reviewer were related to theoretical depth, diversity of the missingness process, and fairness metrics, and we have addressed them in the response below and also in the paper. If you find that these responses are satisfactory, we would be very grateful if you could update your score. Please let us know if you have any further questions.
>
> > Limited theoretical depth ... A more formal analysis of the optimization dynamics would strengthen the paper.
>
> We appreciate the reviewer’s comments regarding the theoretical depth of our work. Our goal in the theory section was to provide principled reasoning for the proposed 3-player min–max framework and to establish its robustness and fairness guarantees under adversarial missingness. We establish several theoretical contributions:
> - Computational hardness: We show that the adversarial missingness problem (AMMBD) is NP-hard (Theorem 1), highlighting its inherent difficulty and motivating the need for approximations and heuristics.
> - Convergence of BFtS: We prove that BFtS yields worst-case fairness imputations (Theorem 2) and demonstrate its robustness to mode collapse and vanishing gradients (Corollary 1), two phenomena known to hinder the convergence of adversarial training and prevent min–max procedures from reaching Nash equilibria [1, 2]. These analysis support the practical stability of BFtS.
> - Fairness guarantees: We establish that BFtS achieves a fairness guarantee: the maximal ΔDP induced by BFtS is minimal among all estimators of the missing sensitive attribute (Theorem 3).
>
> To further examine the convergence, we have plotted the training loss curves for $\mathcal{L}\_{class}$, $\mathcal{L}\_{bias}$, and $\mathcal{L}\_{imp}$ in Figure 6 of the Appendix.
>
> A full convergence or generalization analysis of the 3-player objective is beyond the scope of this work (we mainly focus on the adv. missingness process and the application of BFtS). We agree that a deeper study of the optimization dynamics would be valuable, and we plan to pursue this in future work.
>
> > Comparison to related methods. …how its mechanism differs from other fairness-aware imputation... without deeper discussion.
>
> We have added the following to paragraph at line 466:
>
> “We group the baselines into two categories. First, we compare with methods that are explicitly designed to handle missing values: Debias, FairGNN, and RNF. Among these, RNF can operate when all sensitive attributes are missing, whereas Debias and FairGNN require access to at least some sensitive information. Both RNF and FairGNN rely on independent sensitive-attribute imputation procedures. Second, we compare our approach with state-of-the-art fair GNN models, including FairSIN and FairVGNN. These methods cannot accommodate missing values directly, so we apply an independent imputation strategy consistent with the procedure used in FairGNN to supply the missing sensitive attributes before training.”
>
> BFtS outperforms the baselines because it effectively implements the worst-case assumption for missing value imputation. This assumption leads to stronger fairness guarantees (Theorem 3)  than the baselines in cases where missing values are adversarial. We have added this at line 524. We have also added the following statement from the Appendix to the main text at line 426. `As $f_{class}$ is a minimax estimator, the maximal $\Delta DP$ of BFtS is minimum amongst all estimators of $s$.’
>
> > Experimental diversity…Including other structural or attribute-based missingness patterns ... empirical evaluation more convincing.
>
> In Fig. 2, we previously showed results for two missingness processes, Missing Completely At Random (MCAR) and a degree-based adversarial one. The results show that the bias associated with degree-based missingness is higher than for MCAR, which is because low-degree nodes are more vulnerable to attacks. In Fig. 2 we have added similar results for a new missingness process ‘targeted’ where we select missing values from a candidate set $\mathcal{V}\_{miss} = ${$v \in \mathcal{V}| (y\_v = 1 \wedge s\_v = 0) \vee \ (y\_v = 0 \wedge s\_v = 1)$}. We have added the details at line 256. Among the three missingness processes, the degree heuristic exhibits the largest bias discrepancy between imputed and original values (i.e., is more adversarial).
>
> > Fairness metrics and discussion … why these particular measures were selected and whether the method generalizes to others.
>
> Our theoretical analysis guarantees Demographic Parity, but we can change the $\mathcal{L}\_{bias}$ loss to account for a different type of fairness [3]. In Tables 3 and 4 in Appendix A, we have added results of Equalized Odds [4] (another popular group fairness metric) for all datasets and baselines. We have added these explanations at line 862 in the Appendix.

---

> ### Author Response · Authors · 2025-11-21
> **Continued rebuttal**
>
> > Presentation details: … could be improved for clarity.
>
> We have rechecked the equations and fixed minor inconsistencies in Equations 1 $(\mathcal{V}\_L)$ and 2 $(\mathcal{V}\_S)$ and in lines 357 and 365. We have also improved the following figures: 1 (larger fonts), 2 (larger fonts), 5(larger marker), and 6(larger marker).  We will be happy to also address additional issues in the notation and figures found by the reviewer.
>
> [1] Zhang, Zhaoyu, Changwei Luo, and Jun Yu. "Towards the gradient vanishing, divergence mismatching and mode collapse of generative adversarial nets." CIKM 2019.
>
> [2] Kreps, David M. "Nash equilibrium." The new Palgrave dictionary of economics. Palgrave Macmillan, London, 2018. 9251-9258.
>
> [3] Madras, David, et al. "Learning adversarially fair and transferable representations." ICML, 2018
>
> [4] Grant, David Gray. "Equalized odds is a requirement of algorithmic fairness." Synthese 2023.

---

### Author Response · Authors · 2025-12-02
**Summary of the rebuttal**

Dear Area Chair,

Due to the change in the discussion process for ICLR this year, we want to summarize how our rebuttal addresses the main concerns raised in the reviews.

All reviewers have rated our paper as *good* in terms of *soundness* and *contribution*, while reviewers 8hmM and 53p8 rated our paper’s *presentation* as *fair*. Thus, we have focused on clarifying multiple aspects of our paper (motivation, notation, assumptions, theory, solution, results, etc.) to improve presentation. In particular, we have clarified our theoretical contributions (1oQi), motivated the adversarial missingness process (8hmM), and better supported our use of the degree bias assumption even in heterophilic settings (53p8).

Moreover, we have improved our experiments by adding results using a new adversarial attack (1oQi), increasing the range of missing value ratios in Figure 8  (8hmM, 53p8), and incorporating results using Equalized Odds as an evaluation metric (53p8).

Ideally, we wish we had the opportunity to hear and respond to the feedback from the reviewers. Given the new circumstances, we hope that the AC will be able to assess how our rebuttal addresses the main weaknesses and questions listed in the reviews.

Best regards,

The authors of Submission22075

---

### Meta-Review · Area_Chair_BR3Z · 2025-12-03

**Summary:**

This paper investigates graph fairness when demographic attributes are partially observed under adversarial missingness processes. Based on a thorough review of the reviewers' feedback and my own reading of the paper, I find the reviewers' evaluation to be fair. The authors have clarified multiple aspects of this paper (motivation, notation, assumptions, theory, solution, results, etc.) to improve presentation. Overall, I recommend accepting this paper.

**Reviewer Concerns:**

The authors have clarified theoretical contributions, motivated the adversarial missingness process, and better supported our use of the degree bias assumption even in heterophilic settings.

**Reviewer Scores:**

The authors have improved experiments by adding results using a new adversarial attack (1oQi), increasing the range of missing value ratios in Figure 8 (8hmM, 53p8), and incorporating results using Equalized Odds as an evaluation metric (53p8).

---

### Decision · Program_Chairs · 2026-01-26

Accept (Poster)